# Bibliometric Analysis of IoT Lightweight Cryptography

Zenith Dewamuni †, Bharanidharan Shanmugam *,†, Sami Azam † and Suresh Thennadil †

Energy and Resources Institute, Faculty of Science and Technology, Charles Darwin University, Darwin 0810, Australia; zenith.dewamuni@students.cdu.edu.au (Z.D.); sami.azam@cdu.edu.au (S.A.); suresh.thennadil@cdu.edu.au (S.T.)

* Correspondence: bharanidharan.shanmugam@cdu.edu.au
† These authors contributed equally to this work.

**Abstract:** In the rapidly developing world of the Internet of Things (IoT), data security has become increasingly important since massive personal data are collected. IoT devices have resource constraints, which makes traditional cryptographic algorithms ineffective for securing IoT devices. To overcome resource limitations, lightweight cryptographic algorithms are needed. To identify research trends and patterns in IoT security, it is crucial to analyze existing works, keywords, authors, journals, and citations. We conducted a bibliometric analysis using performance mapping, science mapping, and enrichment techniques to collect the necessary information. Our analysis included 979 Scopus articles, 214 WOS articles, and 144 IEEE Xplore articles published during 2015–2023, and duplicates were removed. We analyzed and visualized the bibliometric data using R version 4.3.1, VOSviewer version 1.6.19, and the bibliometrix library. We discovered that India is the leading country for this type of research. Archarya and Bansod are the most relevant authors; lightweight cryptography and cryptography are the most relevant terms; and *IEEE Access* is the most significant journal. Research on lightweight cryptographic algorithms for IoT devices (Raspberry Pi) has been identified as an important area for future research.

**Keywords:** bibliometric analysis; lightweight cipher algorithm; IoT security

## 1. Introduction

An Internet of Things (IoT) device is a concept with an on-and-off switch for connecting to the internet [1]. IoT is a concept in development in which sensors and smart devices are used to collect and process real-time information from the environment [2]. Over the past twenty years, IoT applications have shown significant growth [3]. IoT technology has many applications, including smart cities, healthcare, education, manufacturing, defense, energy, agriculture [4,5], smart parking systems, smart data acquisition systems, intelligent power grids, and smart data loggers [6].

In light of the increasing number of cyber criminals attacking IoT technology, the security of IoT devices has become increasingly crucial [7]. IoT provides devices with the opportunity to share data among themselves, and these data contain a great deal of confidential information that must be protected [8]. For IoT systems to be protected against unauthorized access, smart cities need high-level security [9]. Traffic light attacks or IoT-based electricity billing attacks can have a significant impact on the day-to-day lives of city residents and infrastructure. To prevent misuse of their data, these IoT systems need high-quality encryption systems [10].

As IoT devices are limited in power, processing, and memory [11], existing encryption algorithms cannot be used in IoT devices. The encryption algorithms used by IoT devices need to be power-efficient, as well as use low processing power and low memory, in order to operate securely [12]. Most IoT devices rely on battery power for power, so these algorithms should not interfere with the device's operation and should not consume excessive amounts of energy for encryption [13]. The National Institute of Standards and

Technology (NIST) has been significantly involved in the standardization of lightweight cryptographic algorithms and they called applications for the standardization process in 2019. A total of ten applications were advanced to the final round and the ASCON family algorithms were announced as the winners in March of 2023 [14]. Several lightweight weight algorithms exist [15], but they need to be tested and validated before being used.

### 1.1. Motivation and Research Questions

After reviewing recent research, it became clear that no bibliometric studies had been conducted on IoT lightweight cryptography. However, bibliometric reviews have been conducted for many areas related to IoT, including smart cities and IoT [16], bibliometric reviews of IoT since its foundation [17], smart homes and IoT [18], and IoT in healthcare [19]. Lightweight cryptography is a key topic of the IoT research area, and authors have identified a lack of bibliometric analysis of IoT lightweight cryptography and information and research gaps. It has provided motivation for authors to conduct bibliometric analyses of IoT lightweight cryptography. In this process, key bibliometric data are identified, facilitating future research on the topic. Therefore, this study was conducted to fill the gap in the literature. In addition to gaining insights into the field's current state, the results of this research will identify possible future areas of research. To accomplish this goal, it is necessary to answer several basic questions. So, this explains the list of questions required in lightweight cryptography (LWC) to prepare the analysis.

- Which countries are contributing the most to the development of IoT lightweight cryptography research?
- Are there any authors who are conducting more research on this topic than the others?
- Does this topic pertain to any trending topics? If so, what are they?
- Which are the leading journals that publish research on this topic?
- What type of co-relations exists between researchers in terms of their research?
- What are the relationships between keywords used in articles?

We will conduct bibliometric analyses using the methods discussed in the Methods section to answer these questions. A significant gap in IoT LWC's bibliometric information is expected to be filled by this research.

### 1.2. Raspberry Pi

Raspberry Pi is a small computer board introduced in 2012 that works like a regular PC and is cheap, powerful, hackable, and geared toward education [20]. Basically, it is a small, credit-card-sized computer that is affordable and able to perform a wide variety of tasks [21], and it can interface with many other devices [22]. Developers can program and develop various applications, such as video games and animations, using graphical programming languages such as Scratch. A programming language such as Python can also be used to develop applications [23]. Raspberry Pi boards provide a number of very good interfaces, including HDMI, Ethernet, USB 2.0, USB 3.0, WLAN, a CSI camera port, and a DSI display port. Boards such as Raspberry Pi 4B have a 64-bit quad-core Cortex A72 mCPU and 4 GB LDDR4 RAM, which gives evidence of the processing capabilities of Raspberry devices [24]. In comparison with other boards, such as Beagle boards, Netduino, and Arduino [22], these capabilities are higher.

### 1.3. Challenges of IoT Lightweight Cryptography Algorithms

Lightweight encryption plays a crucial role in Raspberry Pi data communication and storage [25]. To choose a suitable lightweight encryption algorithm for the Raspberry Pi system, factors such as energy consumption, RAM usage, and execution time should be taken into account [26]. Due to its market share, affordability, ease of use, and wide range of applications, Raspberry Pi has been chosen as the implementation platform for the newly developed lightweight cryptographic authentication in the future. In the IoT, low-resource devices can communicate, process data, and make decisions within the communication network. IoT devices face a number of challenges and issues, including power

consumption, battery life, memory space, performance cost, and security in information Communication Technology (ICT) networks [27]. It will be a challenge for the industry to develop lightweight encryption that meets the above criteria. It will be necessary to fine-tune lightweight encryption algorithms in order to develop micro-size IoT devices that are secure. In comparison with traditional cryptographic algorithms, lightweight algorithms are less resource- and power-intensive [28]. Due to IoT devices' resource limitations, traditional algorithms cannot be used to encrypt IoT data because of hardware damage, instability, and time limitations. Lightweight encryption, however, is more suitable for securing data on IoT devices due to its ability to endure limitations.

## 2. Review of Closely Related Works

A number of review articles have been published on IoT lightweight cryptography. Several of these articles provide an overview of IoT technology, IoT security, challenges associated with using conventional cryptography for IoT devices with limited resources, and an analysis of suitable LWC currently available. An overview of some of the most significant review articles relating directly to this topic can be found in this section.

The architecture and threats associated with IoT, devices at different layers, and securing IoT systems, as well as limitations of existing lightweight cryptography, have been discussed by Rana et al. [29]. They have addressed several questions, including what lightweight cryptography has developed to address security issues in IoT, how lightweight cryptography secures IoT devices, and the implications of the findings for future research in IoT. A series of IoT layers, as well as their vulnerability, were discussed in broad terms in the article. As the title suggests, it summarizes the capabilities of computer boards used in IoT devices as a whole. In addition, an excellent discussion was provided on the cipher methods and algorithms used in lightweight cryptography. The limitations of existing lightweight cryptography for IoT were discussed well. This article is comprehensive and provides relevant information for developing efficient encryption for further research.

Shahzad et al. [30] outlined some key topics related to attribute-based encryption, cipher text, and key management policy in IoT application security. Additionally, they discussed identity-based encryption, searchable encryption, and predicate encryption. A discussion of functional-based encryption is presented in the final section of the article. In addition, they provided excellent illustrations of some applications of encryptions in the form of graphical and tabular representations. Therefore, it provides a comprehensive discussion of all types of encryptions, as well as their limitations and applications. This article provides valuable insight into the process of developing a new cipher algorithm by a researcher.

Mrabet et al. [31] explored IoT technologies, IoT architecture, IoT physical devices and sensors, IoT communication and network protocols, IoT application layers, transport layers, and cloud services. Additionally, they discussed the lack of a standardized lightweight encryption standard for IoT devices and the use of machine learning to enhance the security of IoT devices. As a final point, they discussed the security of blockchain systems and cellular networks. The article discusses gaps in current IoT security technology, which provides an indication of areas that need further research.

Harbi et al. [32] provided an in-depth discussion of current trends in IoT security. At the outset, they discussed IoT devices, IoT architecture, IoT applications, and lessons learned from previous attacks. Threats to different layers of IoT security were also discussed, including those at the perception, network, and application layers. An in-depth discussion of emerging security solutions was conducted, including fog-based solutions, edge computing solutions, software-defined networks-enabled solutions, blockchain-based solutions, lightweight-cryptography-based solutions, homomorphic solutions, searchable solutions, and machine learning-based solutions. The discussion concluded with a discussion of security challenges and future directions. Despite providing a great deal of information regarding modern security solutions, this article does not go into detail regarding each topic.

Thakor et al. [33] focuses on the use of lightweight cryptographic algorithms on devices with limited resources. The article presented a brief overview of IoT, followed by an examination of traditional cryptography algorithms, challenges, and security requirements. After a brief overview of general characteristics, hardware and software performance, and a deeper structure-wise classification of lightweight cryptography, the discussion progressed to a more detailed analysis of LWC. There have been comprehensive analyses and comparisons of a few selected algorithms, and this is an excellent article for understanding existing LWC. The article then goes on to provide an in-depth analysis of the various algorithms, including their strengths and weaknesses, as well as their overall performance. This allows readers to better understand the pros and cons of each algorithm and decide which one is best suited for their specific needs.

Thabit et al. [34] have reviewed cryptography algorithms for enhancing the security of IoT devices. As a prelude to the discussion, a brief overview of IoT and its security was provided. In addition to cyber security and cipher methods, they explored the security trio. There was a detailed discussion of IoT architecture layers accompanied by graphical representations. They also discussed some of the applications of IoT cloud technology. Later in the article, they discuss IoT security challenges in terms of types of attacks, vulnerabilities, threats, and security requirements. They discussed the key challenges associated with using traditional cryptography in IoT devices, as well as lightweight cryptography in areas such as LWC usage, classification, and applications.

Singh et al. [27] described advanced lightweight algorithms for IoT devices. There was a discussion of the ciphers used and the hashes calculated, as well as the performance matrix of a low-performance device and a high-performance device. Asymmetric and symmetric LWCs have been discussed, as well as LWCs for cloud computing and IoT. The final section discusses the hybrid lightweight algorithm (HLA), which is a combination of asymmetric and symmetric algorithms. To illustrate the practical aspects of the subject, some graphical representations and mathematical representations were used. This article is unique among others due to the introduction of hybrid LWC encryption, and it provides good insight for researchers who wish to develop a new LWC. The article concludes by discussing security challenges and countermeasures.

Tao et al. [35] examined the use of hardware for secure data collection in IoT-based healthcare. After providing an overview of IoT technology, the article moved on to discuss security challenges and threat models. As they progress, they discuss how LWC technologies may be used to design secure data. Having discussed the technical aspects of LWC algorithms, the KATAN algorithm [36] was selected. They discussed hardware ciphers, secret cipher share technology, and cipher design using KATAN as a cipher algorithm. An understanding of how to implement the cipher algorithm can be gained by reading this article's implementation and performance analysis.

In Table 1, some of the key points of the eight articles reviewed in this section are summarized. In light of these findings, IoT LWC plays an important role in ensuring the security of IoT devices. Several of the articles discussed the challenges associated with conventional cryptography in IoT devices with limited resources. There is no doubt that LWC algorithms are essential in order to encrypt data on IoT devices. It has been found that there has been no history of bibliometric analysis of the IoT lightweight cryptography problem. The purpose of this article is to examine recent trends in the field of IoT lightweight cryptography, including the nations and authors who were involved, the keywords that were most commonly used, and the collaborations that took place between researchers.

The above article's contents are summarized as follows.

**Table 1.** Summary of existing work.

| Related Work | Source Journal | Year | Objectives |
|:---:|:---:|:---:|:---:|
| [34] | Elsevier Internet of Things | 2023 | Discuss importance of lightweight algorithms and compare cryptographic algorithms |
| [29] | Future Generation Computer Systems | 2022 | Discuss state-of-the-art lightweight protocols, analyze contemporary ciphers, and evaluate recently developed block ciphers and stream ciphers. |
| [30] | Sensors | 2022 | Review functional encryption application and cryptographic primitives |
| [32] | *IEEE Access* | 2021 | Provide up-to-date vision to current IoT security. |
| [33] | *IEEE Access* | 2021 | Compare existing algorithms in terms of some parameters and discuss demand and direction for new research in IoT lightweight cryptography. |
| [31] | Sensors | 2020 | Present open research issues and future directions of IoT security. |
| [35] | IEEE Internet of Things | 2018 | Provide detailed analysis comparing all cryptographic algorithms and their use in day-to-day life. |
| [27] | Ambient intelligence and Humanized Computing | 2017 | Discuss how resource limitations of IoT affect its security and analyze some of the lightweight cipher algorithms and properties. |

## 3. Materials and Methods

### 3.1. Bibliometric Analysis

Bibliographic analysis is a method that locates, compiles, and analyzes metadata in order to show how a field of knowledge has developed over time [37]. A bibliometric analysis is based on the bibliometric data of the data set that was used for the bibliometric analysis. For this analysis, we aim to conduct a performance and scientific mapping of bibliometric information related to the implementation of lightweight cryptography in IoT. Below is a diagram (Figure 1) that illustrates the basic protocol that is used in this research.

### 3.2. Data Collection

The Scopus database was chosen for article searches for the bibliometric analysis since it contains the largest repository and its publications follow peer-review standards [38], which are known for their scientific excellence [39,40]. Web of Science (WOS) has less coverage than Scopus, but its articles are closely related [41,42], which is the reason for selecting it. IEEE Xplore was selected due to its strong coverage of computer and electronic engineering subject areas, but most of the articles were already available with Scopus, which were removed by the data cleansing process.

The article search was run as per the methods mentioned in Appendix A and the identification process of records is shown in Figure 2. Table 2 summarizes the results of article searches.

### 3.3. Data Integration, Cleansing, and Validation

On Scopus and IEEE Xplore, data files were exported as CSV files. WOS generated an Excel file that was later converted to a CSV file, and Microsoft Excel was used to append these files. IEEE Xplore and WOS files were modified to match the Scopus data file. The data cleansing process involved sorting titles alphabetically and removing duplicates as per Appendix B. Using the title as a reference, duplicates were identified using Excel, and the number of articles ended up being 1014. For data integration, cleansing, and text superscript, the R package is also a good option [43], but Excel is much easier to use. Integrating appended files from different databases was challenging as column headers were used with different wording. Manually matching the columns and copying and

pasting data is the most reliable and easy method as the accuracy of the data on the file that we use for analysis is critical. Removing duplicates is important and can be accomplished by following the duplicate-removing process in Appendix B. As an alternative, manual data cleansing can also be performed, but it is a time-consuming process. To validate the data, we manually checked the data to determine whether the articles were related to our analysis, but machine learning can be used for data validation if the data set is too large [44].

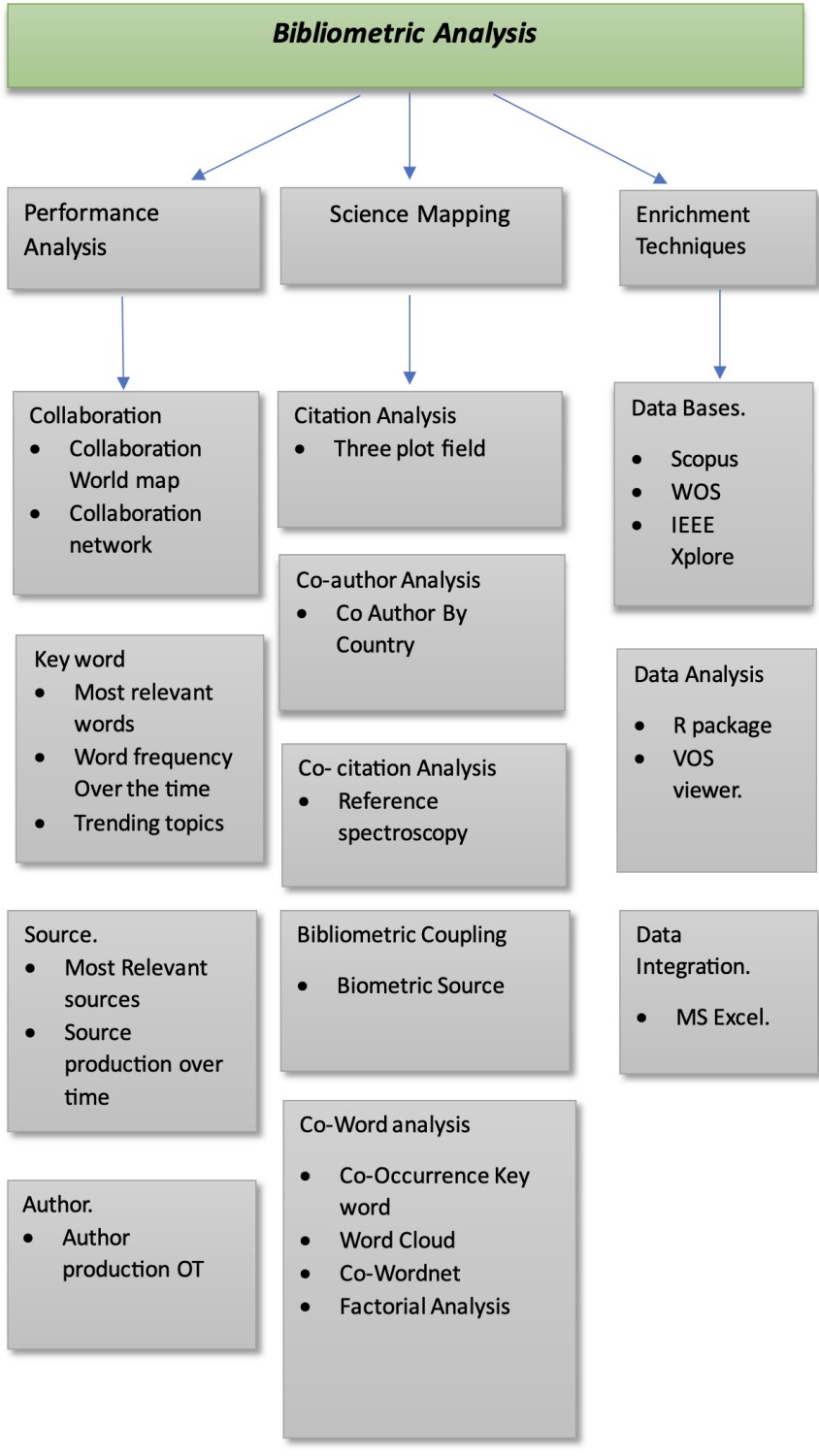

**Figure 1.** Structure of bibliometric analysis.

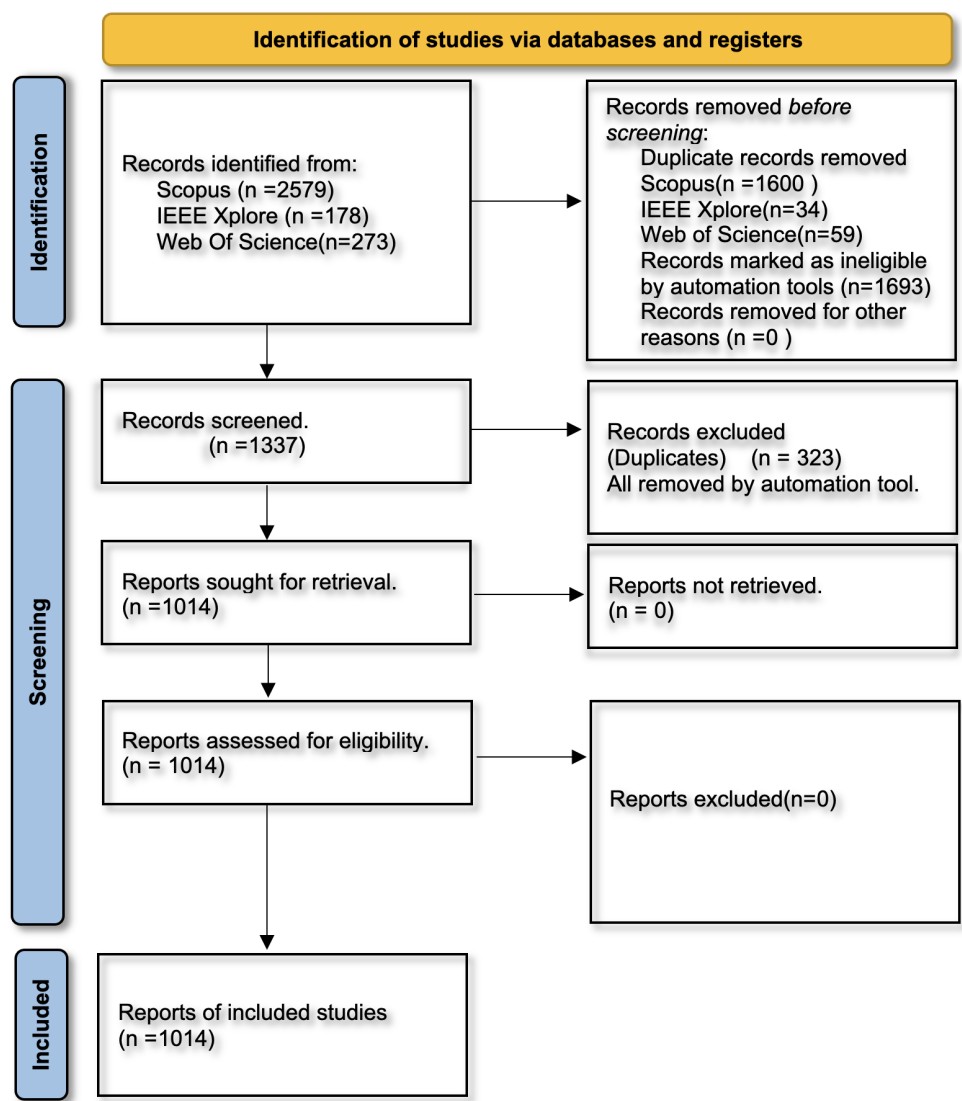

**Figure 2.** Article selection procedure.

## *3.4. Data Analysis*

For analyzing bibliometric data, there are many tools available. For a comprehensive analysis of bibliometric data, the R package is one of the most effective tools [45]. The biblioshiny component of the bibliometrix library was used to carry out the analysis. After choosing raw files as the database type and Scopus as the source file type, the data file was loaded into the data section of the biblioshiny data input section as per Appendix C. To obtain better and clearer results, a related analysis was conducted, and modifications were made, e.g., choosing the required visualization type yielded the required results as per Appendix C. The following types of visualizations were chosen: coupling of countries, co-occurrence of keywords, collaboration world map, most relevant words, word cloud, co-word map, word frequency over time, thematic map, trend topics, cluster by document coupling-word net, factorial analysis, most relevant sources, source production over time, most relevant authors, author production over time, and collaboration net. VOS viewer was also a good tool for analyzing bibliometric data [46], which was used to obtain some results in this analysis by using the steps described in Appendix C. Coupling of countries, bibliometric sources, and co-occurrence of the keyword were some of the results obtained using VOSviewer.

**Table 2.** Summary of article search.

| Database | Search Within | Search Terms | Time Range | Results | Export File Format |
|---|---|---|---|---|---|
| Scopus | Keywords | "IoT" AND "Lightweight Cryptography" | 2015–2023 | 585 | .CSV |
| Scopus | Keywords | "IoT" AND "Light weight Cryptography" | 2015–2023 | 233 | .CSV |
| Scopus | Keywords | "IoT" AND "Light-weight Cryptography" | 2015–2023 | 233 | .CSV |
| Scopus | Keywords | "Internet of Things" AND "Lightweight Cryptography" | 2015–2023 | 816 | .CSV |
| Scopus | Keywords | "Internet of Things" AND "Light weight Cryptography" | 2015–2023 | 357 | .CSV |
| Scopus | Keywords | "Internet of Things" AND "Light-weight Cryptography" | 2015–2023 | 357 | .CSV |
| WOS | Author Keywords | "IoT" AND "Lightweight Cryptography" | 2015–2023 | 134 | .xlsx |
| WOS | Author Keywords | "IoT" AND "Light weight Cryptography" | 2015–2023 | 7 | .xlsx |
| WOS | Author Keywords | "IoT" AND "Light-weight Cryptography" | 2015–2023 | 7 | .xlsx |
| WOS | Author Keywords | "Internet of Things" AND "Lightweight Cryptography" | 2015–2023 | 119 | .xlsx |
| WOS | Author Keywords | "Internet of Things" AND "Light weight Cryptography" | 2015–2023 | 3 | .xlsx |
| WOS | Author Keywords | "Internet of Things" AND "Light-weight Cryptography" | 2015–2023 | 3 | .xlsx |
| IEEE Xplore | Author Keywords | "IoT" AND "Lightweight Cryptography" | 2015–2023 | 77 | .CSV |
| IEEE Xplore | Author Keywords | "IoT" AND "Light weight Cryptography" | 2015–2023 | 6 | .CSV |
| IEEE Xplore | Author Keywords | "IoT" AND "Light-weight Cryptography" | 2015–2023 | 6 | .CSV |
| IEEE Xplore | Author Keywords | "Internet of Things" AND "Lightweight Cryptography" | 2025-2023 | 79 | .CSV |
| IEEE Xplore | Author Keywords | "Internet of Things" AND "Light weight Cryptography" | 2015–2023 | 5 | .CSV |
| IEEE Xplore | Author Keywords | "Internet of Things" AND "Light-weight Cryptography" | 2015–2023 | 5 | .CSV |

## 4. Results

Based on the methodology described above in Section 3, the data were analyzed, and the results are presented graphically and tabulated in this section.

### 4.1. Co-Author by Countries

As shown in Figure 3, India has the highest number of co-authors, followed by China and the United States. These three players play a significant role in terms of the number of co-authors. In addition to these, some other countries, such as France, Japan, the United Kingdom, Canada, Australia, Sweden, Luxembourg, and South Korea, are also part of the first-world developed country segment, which is highly technical and rich in technical resources. It is important to note that these countries are major players in the field of lightweight cryptography research. A number of countries in the Middle East, such as Saudi Arabia, Egypt, Iraq, Pakistan, Turkey, and Iran, have also shown a significant interest in this topic from a research perspective. In this analysis, bigger nodes represent a high volume of articles produced, and thicker lines represent the strength of the research relationships between these countries. There is no doubt that many of the countries in the world are showing good research interest in this topic, and there should be a good demand for this topic in the future. It is useful to analyze Figure 3 to determine which countries are more involved in LWC research. It would be helpful if future researchers understood which countries conduct more research in certain areas. Additionally, it serves as a knowledge database, as well as providing insight into collaborations. Additionally, entrepreneurs and technology investors will have a better understanding of which countries are leading the research, enabling them to make better business decisions.

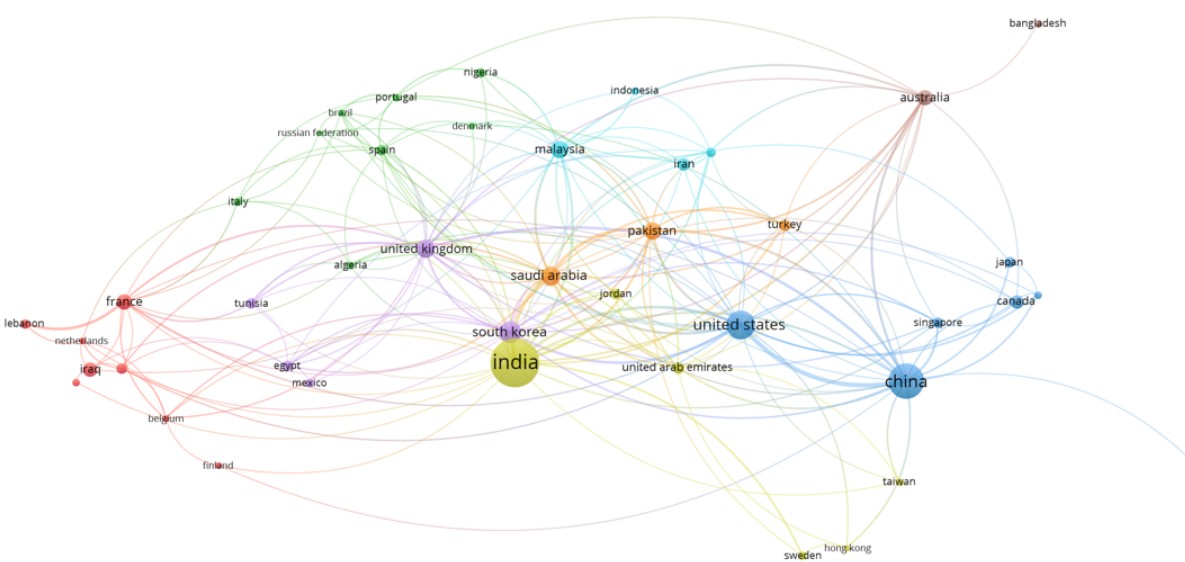

**Figure 3.** Co-Author by Country.

### 4.2. Co-Occurrence of Key Words

Figure 4 shows that the Internet of Things, cryptography, and lightweight cryptography are all major keywords, which are also our selection keywords. Besides the main keywords, there are a number of other keywords that are closely related to them. Bigger nodes represent the high volume of appearance, and thicker lines represent higher relationships. There are four different clusters of keywords that have been identified, and they are each represented by a different color. A bunch of keywords related to IoT and cryptography are clustered together in the green cluster and linked to other keywords closely related to that research field. Lightweight cryptography appeared in the blue cluster with closely related topics such as cipher algorithms. Network security and security lead the red cluster. There is a greater relevance of these keywords to the IoT lightweight cryptography application. In the future, it will be a good source to help researchers find the most relevant articles in a particular field. When a few keywords are combined to form a combined search term in a database, it results in a very good search result. Based on the figure above, we can gain valuable insight into the search articles and provide good directions for future research.

### 4.3. Collaboration World Map

Figure 5 shows India is leading the way in terms of contributions to LWC research. In addition to China, the USA, the UK, and South Korea show a great deal of interest in this topic. There are strong ties between India and Saudi Arabia, China and the USA, India and Australia, the USA and Korea, and India and the USA. Research efforts in modern science are often conducted jointly, especially in emerging fields. A clear picture can be drawn of India, the USA, China, the UK, and South Korea collaborating on this IoT LWC research. About one-third of the research is conducted in India and China. These countries will become the main hub for IoT LWC research. Moreover, the USA has a strong research interest in this area with a good level of collaboration with China and India. In LWC, these countries are at the top of the list for researchers who wish to collaborate with other countries. Table 3 shows the percentage of contribution by top 5 countries in terms of collaboration.

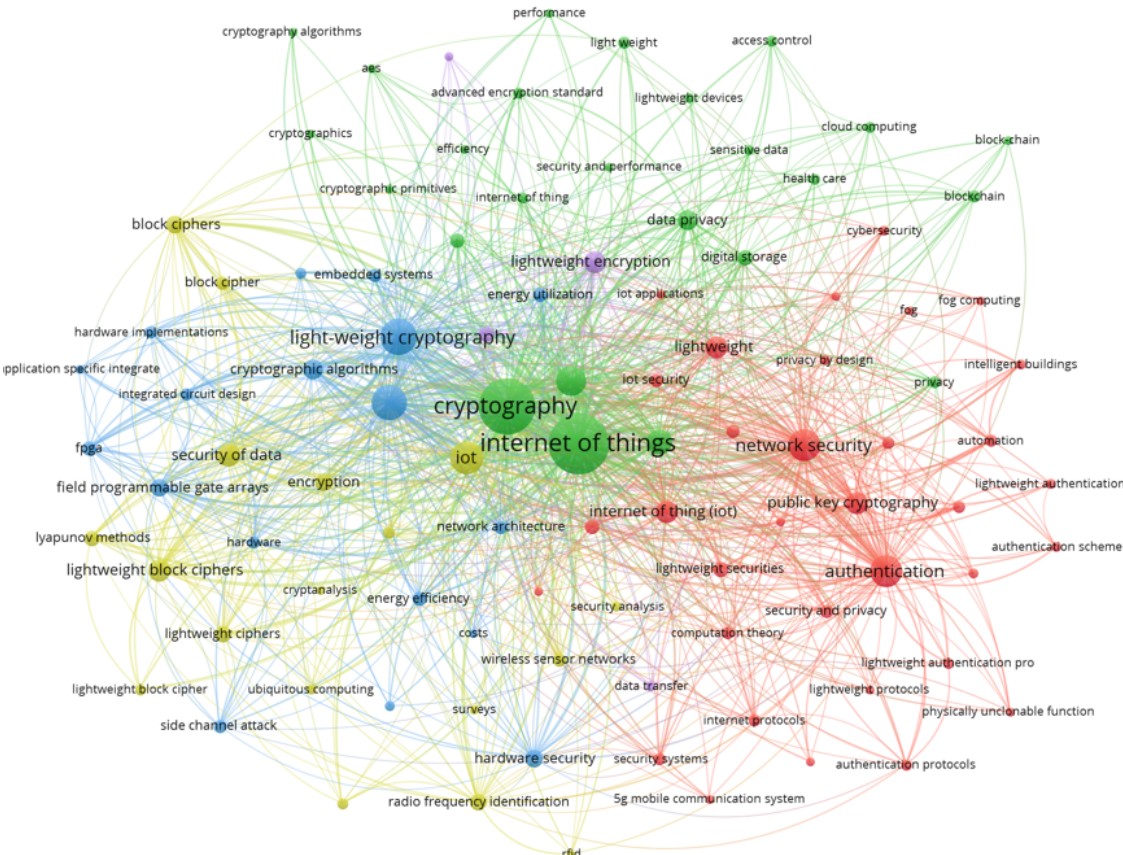

**Figure 4.** Co-occurrence of keywords.

**Table 3.** This table shows the percentage of contribution by the top 5 countries.

| Country | Percentage of Contribution |
|---|---|
| India | 16.9 |
| China | 15.2 |
| USA | 11.0 |
| South Korea | 7.3 |
| United Kingdom | 6.9 |

*4.4. Collaboration Network*

Figure 6 shows how the collaboration network can identify contribution clusters up to ten. The contribution of Archarya to the IoT LWC research area plays a significant role in this research by representing his name in larger text. Table 4 shows the cluster number against the color of the cluster for easy reference. There is no doubt that he has a high page ranking, as is reflected in the larger text that appears next to his name. In the segment represented by the largest node, he has a high degree of closeness with his colleagues. This researcher collaborates mainly with five other researchers in that cluster. Bansod is also a significant player who appears in Cluster 7. Moreover, he has a higher page ranking and shows a stronger connection with other researchers, as evidenced by a larger number of nodes and larger text. He is part of a research cluster that includes four other researchers. Other clusters do not have any leading names, but Cluster 3, which is in light green, has nine researchers around.

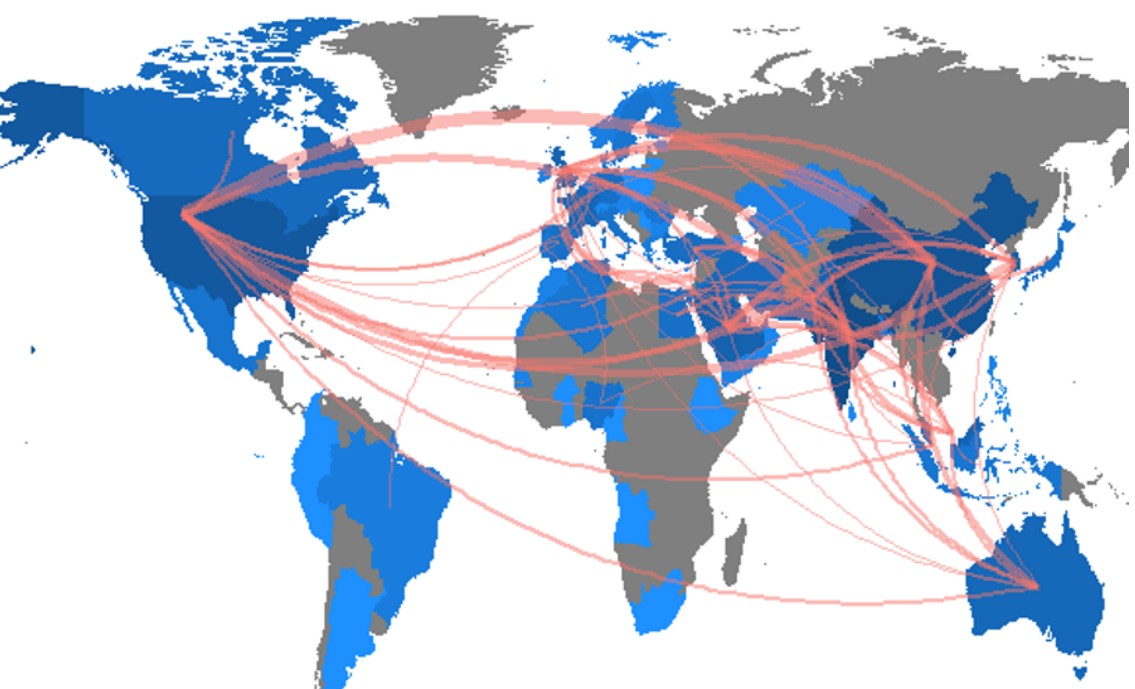

**Figure 5.** Collaboration world map.

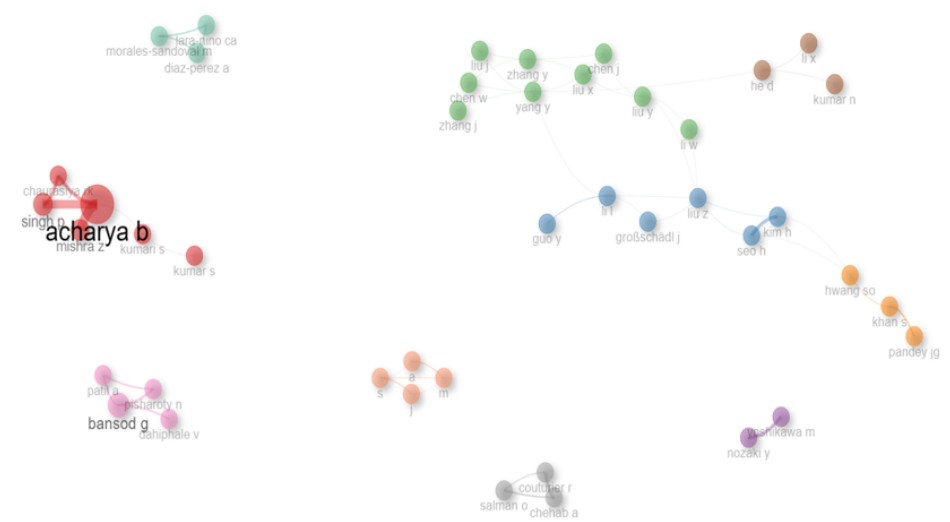

**Figure 6.** Collaboration network.

**Table 4.** Collaboration network data.

| Cluster Number | Color |
| --- | --- |
| 1 | Red |
| 2 | Blue |
| 3 | Light Green |
| 4 | Purple |
| 5 | Orange |
| 6 | Brown |
| 7 | Pink |
| 8 | Gray |
| 9 | Green |
| 10 | Dark orange |

*4.5. Most Relevant Words*

Figure 7 shows how often each keyword appeared in the data set and the number of times each keyword was used. In the data set, the keyword lightweight cryptography appeared 305 times. Internet of Things and IoT appeared 509 times as a combined result. There were 119 instances of cryptography appearing in the most relevant word search and 151 instances of security. Based on our search of our database around these three topics, we can conclude that our data set is more relevant. Researchers using the IoT LWC can use this analysis to determine whether their literature data set is relevant to the topic. Based on the analyzed data set, it is clear that IoT LWCs are closely related. Another word that appeared in our data set was lightweight, followed by authentication, encryption, and FPGA. This means that these keywords are some of the most related to IoT cryptography. As a result, the IoT keyword appears multiple times for different letter combinations. The purpose of this tool is to help researchers design research and research papers related to the IoT LWC field and to help them write them.

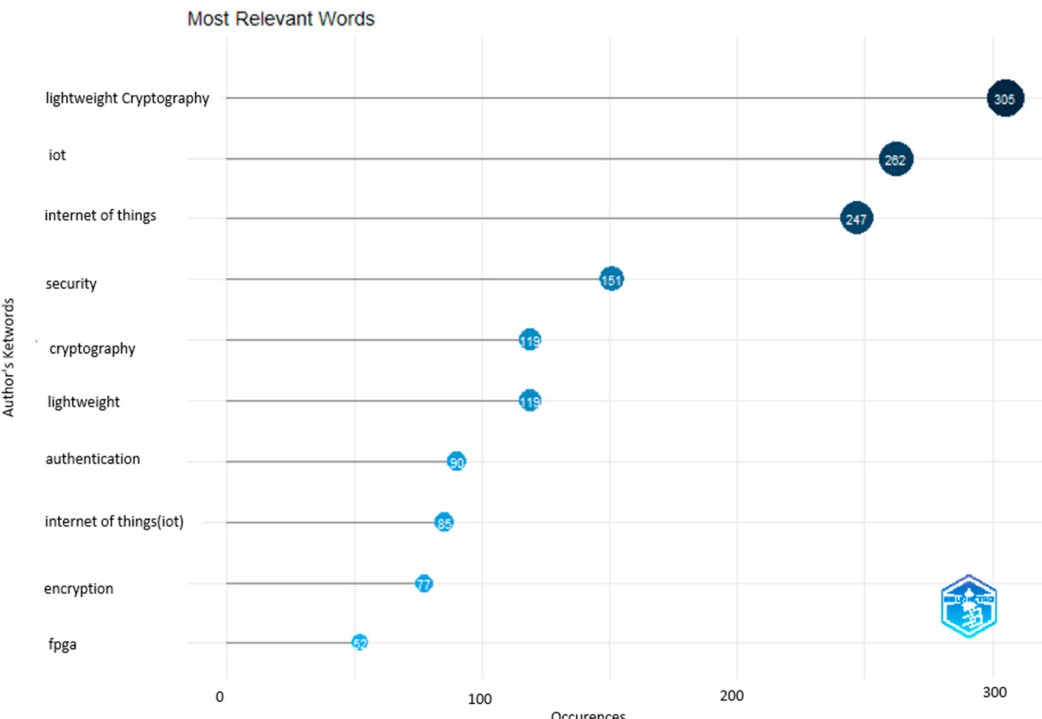

**Figure 7.** Most relevant words.

*4.6. Word Cloud*

In Figure 8, Internet of Things, lightweight cryptography, cryptography, and security are the main texts that appear [47]. Among the analyzed data set, these three keywords are the most prevalent, representing a significant number of articles. The small text also contained other keywords like lightweight, encryption, and authentication, which means that they are closely related to the main keywords [48]. These are partly related topics that could be included in the IoT LWC, and researchers should consider this to get a better idea of how IoT LWC fits into the existing work. In terms of IoT LWC, it is a good tool for setting up research directions and showing the numeric proportion of keywords appearing in articles through the tabular representation of the word cloud. Table 5 shows the number of appearances of the top ten keywords.

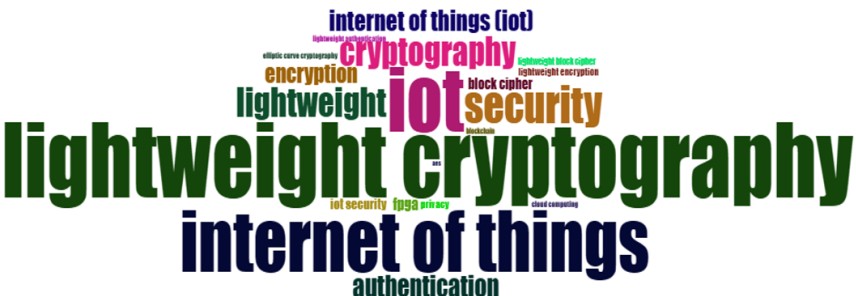

**Figure 8.** Word cloud.

**Table 5.** Top 10 keywords and the number of appearances in the data set.

| Key Word | Number of Appearances |
|---|---|
| Lightweight cryptography | 305 |
| IoT | 262 |
| Internet of Things | 247 |
| Security | 151 |
| Cryptography | 119 |
| Lightweight | 119 |
| Authentication | 90 |
| Internet of Things (IoT) | 85 |
| Encryption | 77 |
| FPGA | 52 |

### 4.7. Co-Word Net

Figure 9 shows that three keywords dominate: Internet of Things (IoT), lightweight cryptography, and security. These are the main topic areas in our search that are directly related to the main keywords we are searching for, and this analysis gives an idea of the co-occurrence of keywords in the data set [49]. The topic is closely related to several other keywords, including encryption and decryption, advanced encryption standards, block ciphers, and hardware security, which are closely related to the main topic, and it gives an idea of how different concepts relate to each other [50]. Bigger nodes represent a higher number of appearances, and thicker lines represent the strong relationship between keywords. There are two clusters that can be identified, but one cluster is dominant, which is highlighted in red. There is a strong correlation between the IoT LWC subject area and that cluster. It is a useful tool for researchers who want to move beyond the major research areas of IoT LWC to develop new research themes or expand existing ones. In Table 6, you can see a table with a list of the top 10 players in the red cluster in a tabular format.

### 4.8. Word Frequency over the Time

Based on the data in Figure 10, it can be seen that keywords related to IoT cryptography and LWC have demonstrated good cumulative occurrences over time. There is no doubt that IoT LWC is a growing research field, which can act as a signal for a new researcher to focus their attention on that area for the sake of a successful research project. In addition, there is another significant piece of information on the graph that indicates that all of these topics started climbing in the graph around 2016, which is an indication that these topics emerged from that time period. In other words, IoT LWC is a relatively new area of research that offers many opportunities for new researchers to improve their skills and understanding of the subject. In Table 7, the performance of the top five keywords over the last 5 years is presented in a tabular format.

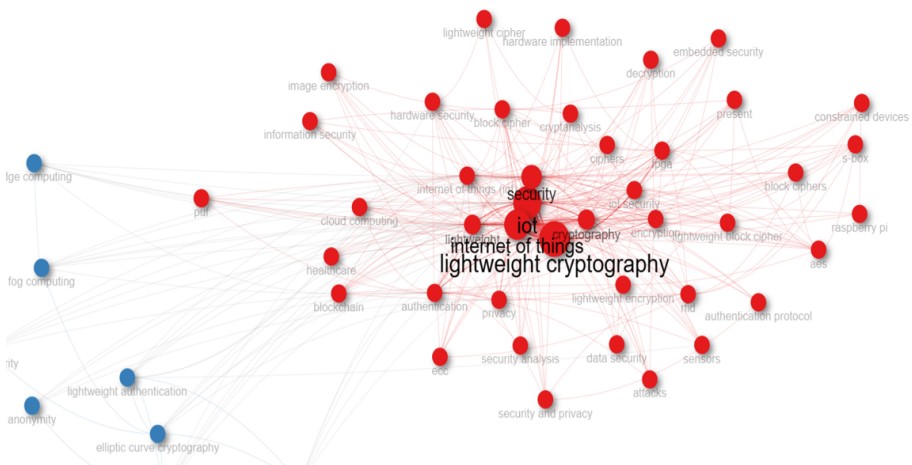

**Figure 9.** Co-Word Net.

**Table 6.** Top 10 keywords and the number of appearances in the data set.

| Key Word | Betweenness | Closeness | Page Rank |
| --- | --- | --- | --- |
| Lightweight cryptography | 222.7456282 | 0.017857143 | 0.113957932 |
| IoT | 225.0987635 | 0.018518519 | 0.106899186 |
| Internet of Things | 237.6984063 | 0.019607843 | 0.095832642 |
| Security | 77.64761793 | 0.016666667 | 0.073906234 |
| Cryptography | 24.65528569 | 0.014084507 | 0.05023756 |
| Lightweight | 41.23171437 | 0.014925373 | 0.049418078 |
| Authentication | 18.67824876 | 0.013513514 | 0.040011909 |
| Internet of Things (IoT) | 16.4028379 | 0.013888889 | 0.029672419 |
| Encryption | 10.32552461 | 0.012987013 | 0.039717612 |
| FPGA | 3.44796444 | 0.012658228 | 0.025962694 |

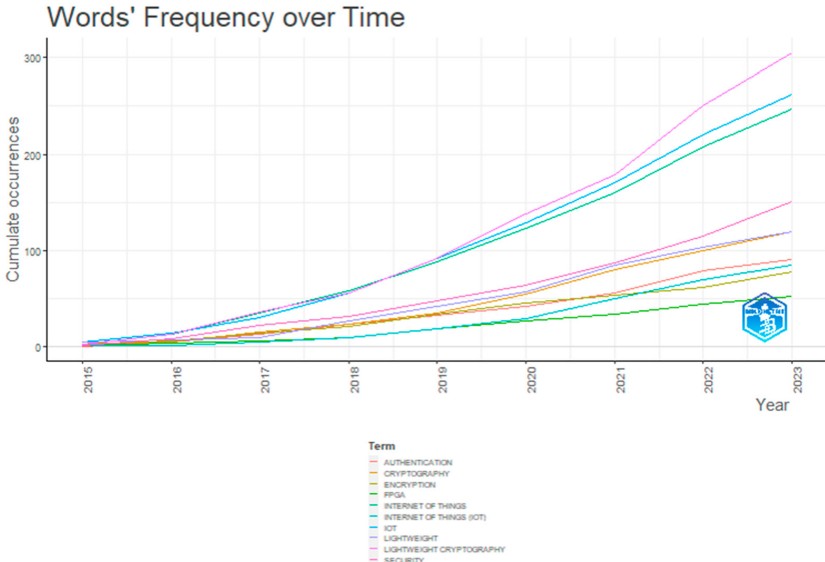

**Figure 10.** Word frequency over time.

**Table 7.** Word frequency over time.

| Year | Internet of Thing/IoT | Cryptography | Lightweight Cryptography/Lightweight | Security | Authentication |
|------|------------------------|--------------|----------------------------------------|----------|----------------|
| 2018 | 113 | 23 | 82 | 31 | 23 |
| 2019 | 179 | 35 | 132 | 47 | 32 |
| 2020 | 252 | 54 | 195 | 64 | 42 |
| 2021 | 330 | 80 | 263 | 87 | 56 |
| 2022 | 471 | 100 | 354 | 115 | 79 |
| 2023 | 509 | 119 | 426 | 151 | 90 |

*4.9. Thematic Map*

Figure 11 shows that cryptography, the Internet of Things, and lightweight cryptography are categorized as basic themes. In terms of basic themes, the development degree is low, but the relevance degree is high. Moving toward the right indicates an upward trend, while moving upward indicates a high degree of trend, and moving toward the left indicates a downward trend [51]. The data set allows us to identify that LWC, IoT, and cryptography are rising topics, but the amount of trending is still low for these topics. There is no doubt that these topics are going to be a trend in the future. Some niche themes can be identified in this analysis, such as the avalanche effect and attribute-based encryption, which are sub-topics of cryptography. Based on the positions found in the graph, it is apparent that those themes are showing a good level of rise and trend over time. There are a number of motor themes that can be identified in the graph, which are still in their rising and trending stages, such as Raspberry Pi, lightweight block cipher, present, and Simon. Post-quantum cryptography is in the emerging themes category, which is in the middle and will be a trending topic in the future.

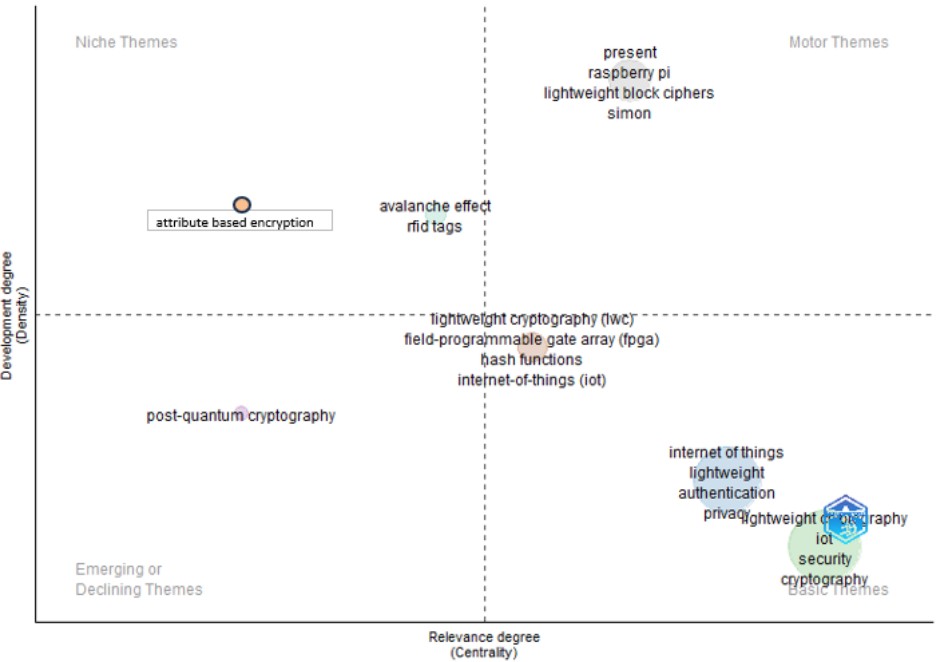

**Figure 11.** Thematic map.

*4.10. Trend Topics*

In Figure 12, a great deal of attention is paid to areas such as IoT, lightweight, security, encryption internet, and algorithms. There are also some emerging topics that can be identified, including RFID, IoT healthcare, elliptic curves, and IoMT. As The keywords usually given to the articles are connected to the publication content, which generates

a topical aspect of the field [52], there is no doubt that this analysis is very useful in understanding the design of the research and where the future focus of the research will be in the research area of IoT LWC as well. A large number of articles dealt with topics such as the Internet of Things, lightweight cryptography, encryption, security, and the internet, which resulted in larger nodes that represented those topics.

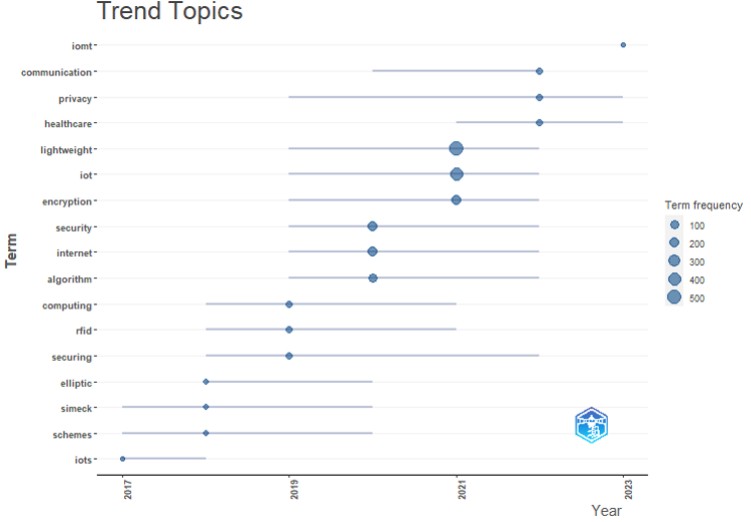

**Figure 12.** Trend topics.

### 4.11. Factorial Analysis

In Figure 13, it can be identified only one cluster, which consists of a number of research topics, in our analysis. This analysis helps us to gain a better understanding of the mature cluster in the nomogram when it comes to the Internet of Things LWC. In this study, only one cluster can be identified as a result of the data analysis, and a number of topics located in the fourth quadrant of the table, such as cryptography algorithms, cryptography, lightweight encryption, and hardware implementations, are better suited for further development. There is no doubt that cryptography and cryptographic algorithms are in that category and it gives a researcher confidence to conduct research in the area of IoT LWC research.

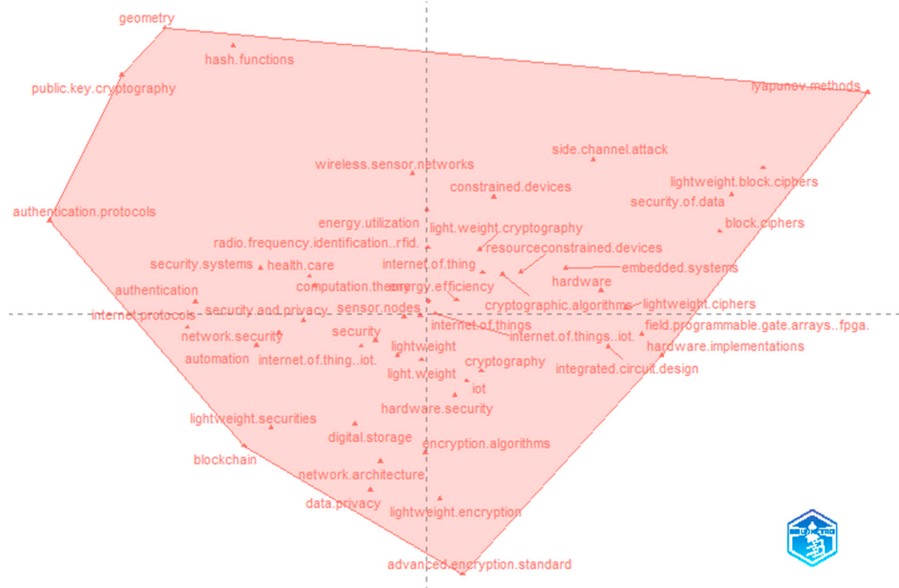

**Figure 13.** Factorial analysis.

*4.12. Bibliometric Source*

Figure 14 illustrates that *IEEE Access* has the highest number of journal articles published in the subject area. There are nodes representing journals, and the bigger the node, the more publications there have been in that journal. *IEEE Internet of Things*, *Lecture Notes in Computer Science*, and *Lecture Notes in Electrical Engineering* are some other top journals. *IEEE Access* and *IEEE Internet of Things* have a good relationship, as indicated by the lines connecting nodes in the graph. There is a good relationship between the lecture notes for computer science and those for electrical engineering. Interestingly, there appears to be a correlation between the thickness of the lines and the strength of the connections between the journals in terms of quality. As a result of this analysis, a researcher will be able to find articles more efficiently and make reference sources more reliable as a result of the analysis. In addition, researchers can also use this method in order to determine which journals to select for publication in particular areas of research.

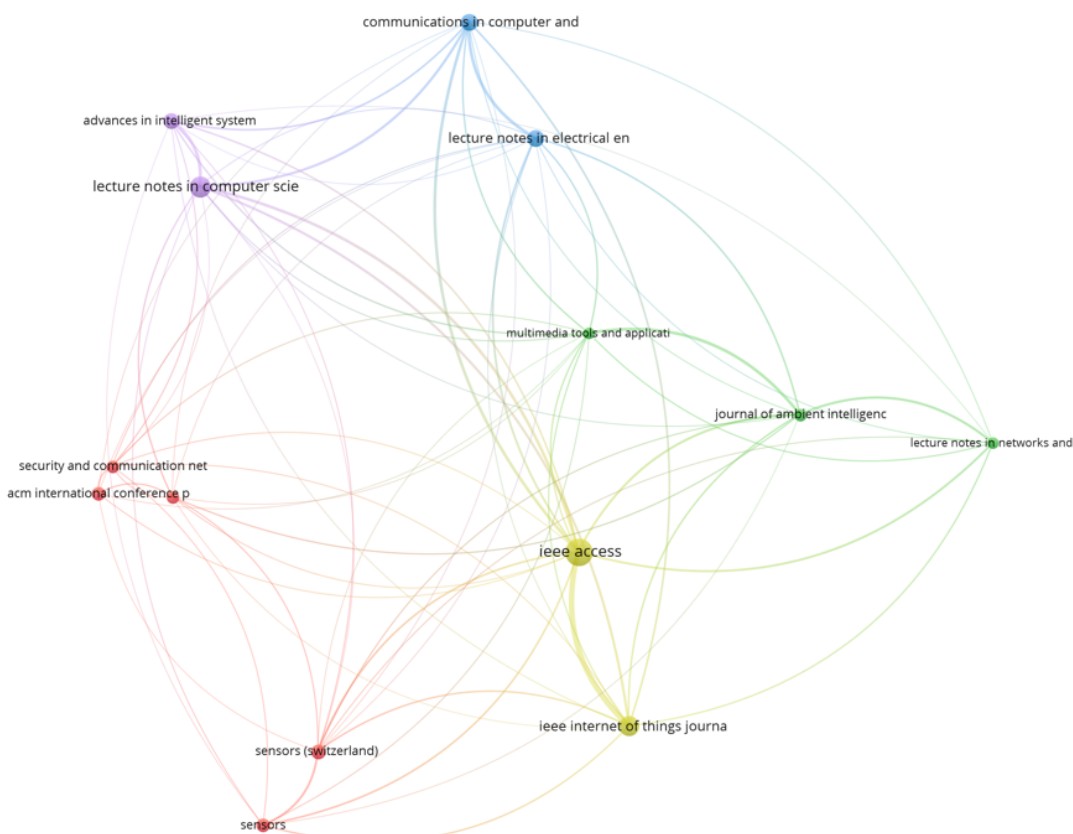

**Figure 14.** Bibliometric source.

*4.13. Most Relevant Sources*

Figure 15 shows that *IEEE Access* is the leading journal in the IoT lightweight cryptography subject area. *IEEE Access* is a free-accessible journal, which may explain its position at the top. *Lecture Notes in Computer Science* and *Lecture Notes in Electrical Engineering* are the second and fourth leading journals published by Springer Nature. *IEEE Internet of Things* is the third leading journal in the IoT LWC area. But *Sensors* appeared two times as *Sensors*, and *Sensors* (Switzerland) is the same MDPI journal, which appeared 31 times together, comes to number four in this analysis. This analysis provides a number of relevant articles that are easy to understand and compare. Analyzing these data allows a researcher to identify relevant articles and references, as well as suitable journals for publishing papers. In terms of publication of articles and searching for references, this diagram will help to get an idea.

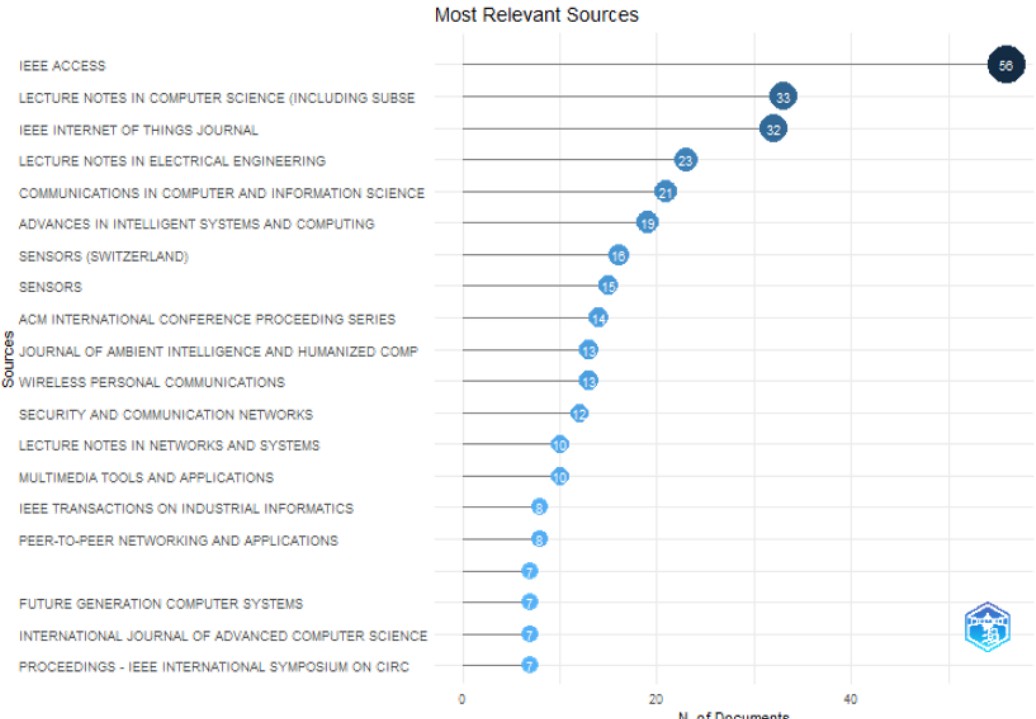

**Figure 15.** Most relevant source.

*4.14. Source Production over Time*

In Figure 16, it can be seen that *IEEE Access* is the most prolific journal in terms of source production over time, and it clearly shows healthy growth in recent years. *Lecture Notes in Computer Science*, *Lecture Notes in Electronic Engineering*, and *IEEE Internet of Things* are some of the other leading journals that have shown similar kinds of growth over the years. However, compared to other journals, the Advanced in Intelligence and Systems journal stopped being active after 2019 compared to other journals. Lecture Notes in Computer Science also shows a plat line during the last couple of years means that the journal is also getting less active. *IEEE Access*, *Lecture Notes in Electronic Engineering*, *IEEE Internet of Things*, and *IEEE Lecture Notes in Electronic Engineering* are the most suitable journals in which to publish or search for references. Table 8 shows the top 5 journals' performances during 2018–2023.

*4.15. Most Relevant Author/Author Production over Time*

Figure 17 clearly shows Archarya and Bangsod are two of the leading researchers in LWC, and any researcher who is interested in research in the IoT LWC field can look at the above list of names and select some articles from it that they may find useful in their research. Table 9 is a tabular representation of the top 10 authors based on their relevance in terms of their most relevant publications. For an accurate assessment of the quality of the authors, it would be better to combine the findings of this analysis with a number of citations from Figure 18 in order to obtain a good idea of their research. As shown in Figure 18, Archarya started a bit later in this field compared to other researchers, but he has consistently contributed to the research field of IoT LWC and is currently active in this field. Mizra, Khan, and Singh are some other similar kinds of researchers. Bansod is an early starter in the IoT LWC research field but has not been active since 2019. Li is another consistent researcher in the IoT LWC research area. The con of the above analysis is that it does not provide any indication of the quality of the articles that have been produced as a result. It is highly recommended to use this analysis in conjunction with a matrix of quality indicators, such as the number of citations that are indicative of the quality of the publication, as it highly depends on the number of publications.

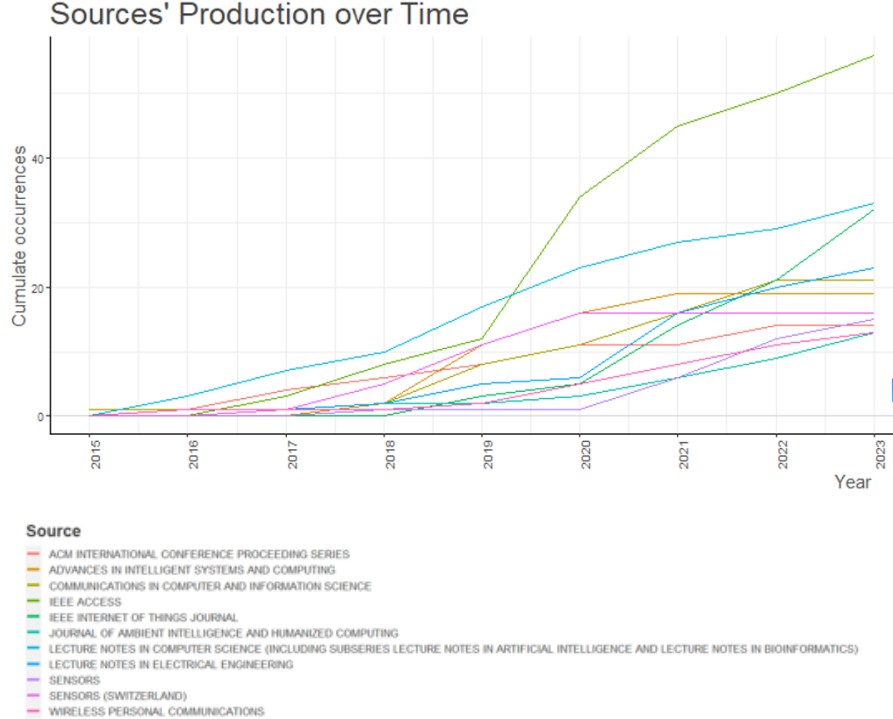

**Figure 16.** Source production over time.

**Table 8.** Cumulative source production over time.

| Year | *IEEE Access* | Lecture Notes in Computer Science | Lecture Notes on Electronic Engineering | IEEE Internet of Things | Communications in Computer and Information Science |
|------|------|------|------|------|------|
| 2018 | 8 | 0 | 2 | 0 | 2 |
| 2019 | 12 | 17 | 5 | 3 | 8 |
| 2020 | 34 | 23 | 6 | 5 | 11 |
| 2021 | 45 | 27 | 16 | 14 | 16 |
| 2022 | 50 | 29 | 20 | 21 | 21 |
| 2023 | 56 | 33 | 23 | 32 | 21 |

**Table 9.** Top 10 most relevant authors.

| Author | Number of Articles | Articles Fractionalized |
|--------|------|------|
| Acharya, B. | 25 | 8.91 |
| Bansod, G. | 15 | 5.6 |
| Li, L. | 12 | 3.65 |
| Mishra, Z. | 12 | 4.6 |
| Yang, Y. | 12 | 2.82 |
| Roy, S. | 11 | 2.43 |
| Singh, P. | 11 | 3.25 |
| Khan, S. J. | 10 | 2.16 |
| Kim, H. | 9 | 2.25 |
| Chehab, A. | 8 | 2.1 |

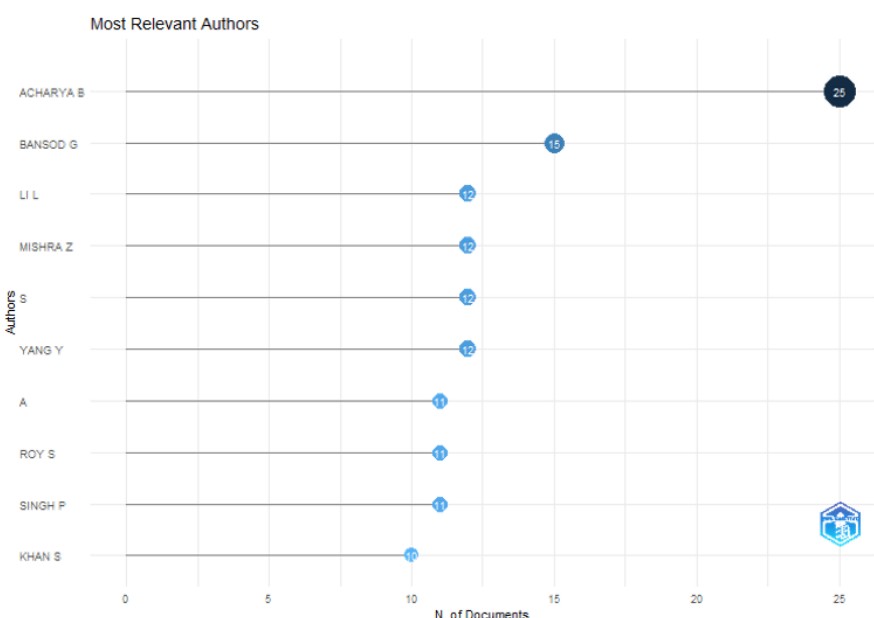

**Figure 17.** Most relevant authors.

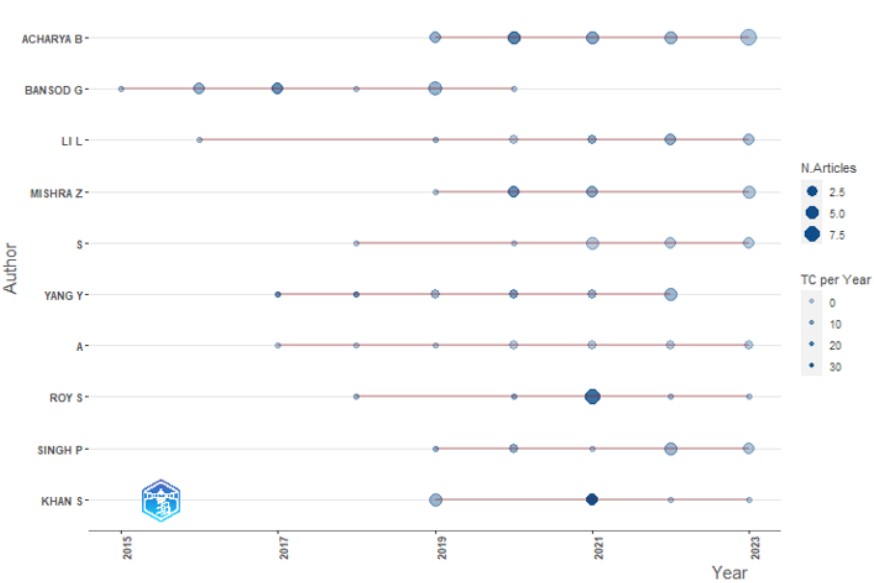

**Figure 18.** Author production over time.

### 4.16. Publication and Citation

Figure 19 illustrates the period that runs from 1975 to 2022. Some of the old articles were also referred to in this data set in order to provide references. In the period from 1990 to 2020, the period of the information age is characterized by the highest number of peaks. If a researcher is looking for articles to use as references, they can narrow down their search to articles published during this period. The number of cited references is represented in blackline and deviation from the 5-year median is represented in redline.

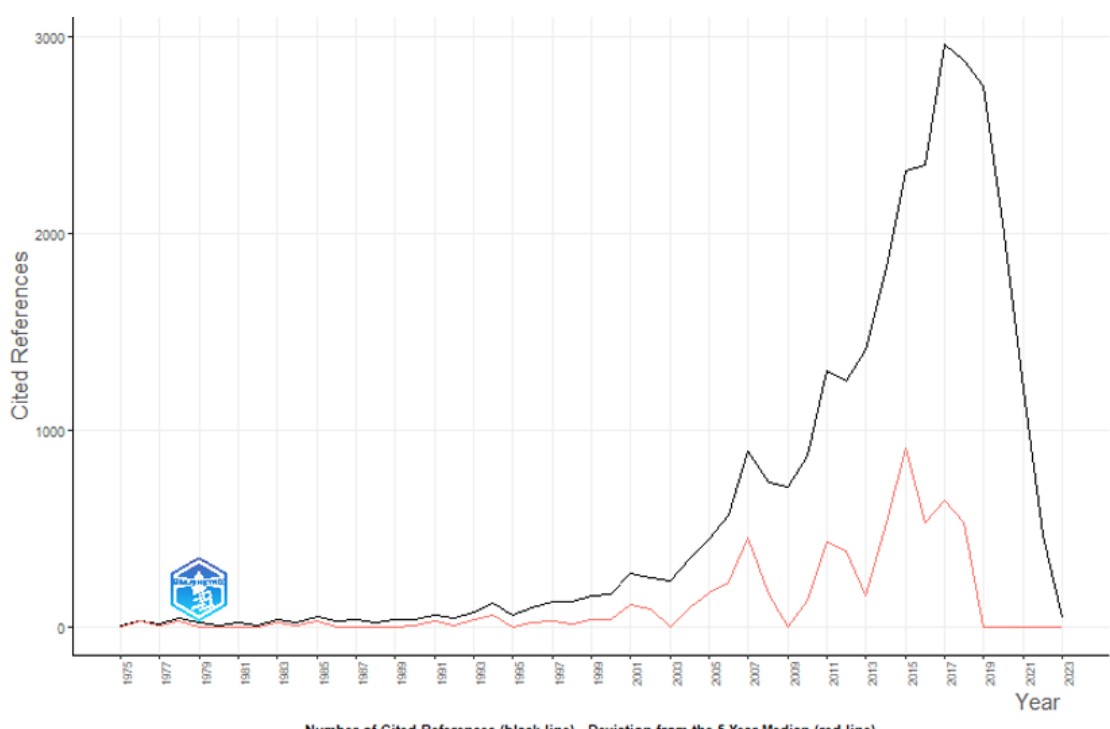

**Figure 19.** Publication and citations.

## 5. Findings

Our goal is to understand the role of lightweight cryptography in IoT devices. In the current analysis, IoT, cryptography, and lightweight cryptography have shown a significant level of appearance. This shows the importance of LWC in IoT technology. Increasing the term frequency, keyword appearance, and co-occurrence of keywords are some of the indications that research in LWC is an emerging field of research. It is an ideal opportunity for a researcher to conduct research in this subject area.

A closer look at the collaboration world map and co-authorship by country show that India leads the research in IoT lightweight cryptography. Some of the leading researchers are based in India and have enthusiasm for technological studies with students. Some institutes' high interest in IoT lightweight cryptography could be the reason behind its success. For researchers and investors seeking technological collaboration, India can be a good choice. In addition to China, the US, and the UK, South Korea, some Middle Eastern countries have shown interest in IoT lightweight cryptography. The research area of IoT LWC is receiving significant attention from all continents, except Africa and Eastern Europe. In that sense, it is a good sign that a new researcher understands the importance of this field.

A keyword analysis shows that IoT, cryptography, and lightweight cryptography are some of the most common words. That can be seen in word clouds and co-word networks. The same number of articles discussed these keywords, which indicates that they were discussed extensively. Over time, lightweight cryptography has shown significant increases in word frequency and trend topics. As a result, lightweight cryptography will become an important research area in the future.

Using author-related analyses such as the most relevant authors and authors' productions over time, we can identify who is the leading researcher around LWC for IoT. Authors such as Archarya and Bansod have conducted several research studies in this area of study. It is evident that the author is continuing to work on their work over the course of time. Archarya is a researcher from the National Institute of Raipur and is working with a network of other researchers. As per the Scopus database, he has produced 118 articles, and his articles have been cited over 1100 times by 722 documents, making significant contributions to the industry. He appeared in our search 25 times, and lightweight cryptography is one of

his major research fields. Bansod has also produced 20 articles with over 500 citations, and his major research fields are embedded systems, low-power cryptography, and lightweight cryptography.

There is no doubt that *IEEE Access* is the leading journal based on an analysis of journals. As per its cover page, it is a multidisciplinary, open-access, and rapid-review journal, which could be the reason behind its success of publishing in the IoT lightweight cryptography subject area. It has an impact factor of 4.82, and it is in the Q1 zone in the SCimago journal ranking with a score of 0.926. The lecture notes on electrical engineering and the lecture notes on computer science and *IEEE Internet of Things* and *Sensors* are also some of the strongest journals in the IoT LWC. This means that these are good journals to find information on IoT LWC for researchers who are just starting out. It is clear from these analyses that researchers will be able to identify which journals are suitable for publishing their articles in the IoT LWC based on these analyses.

These bibliometric analysis results indicate that there are several areas of future research to explore. With topics such as IoT security and IoT applications gaining prominence in modern times, IoT research is a major trend now, and many researchers are now focusing on IoT technology and IoT applications. However, niche areas of research such as IoT lightweight cryptography, which is a hot topic now, are the future trends, as most researchers are focusing on this area in depth. Researchers can continue to narrow down their research topic to areas such as cipher algorithms, block ciphers, and so on if they wish to go further and be more specific with their research.

These results provide a sound understanding of the importance of IoT cryptography as a solid foundation for IoT security. It provides a very comprehensive foundation for a decision-making tool for anyone interested in IoT lightweight cryptography.

*5.1. Answers for Research Questions*

Some of the research questions were set up at the beginning of the article with the intention to answer them based on results obtained from the analysis.

- Which countries are contributing the most to the development of IoT lightweight cryptography research?

Collaboration world map and co-author by country are key analyses to answer this question. Both analyses show that India is the leading collaborator of IoT LWC researchers. China is the second largest country, and the USA, the UK, and South Korea are some other contributors to the IoT LWC research field. There are many middle-level players such as Saudi Arabia, Pakistan, France, Australia, Malaysia, Turkey, and Iran identified by the analysis.

- Are there any authors who are conducting more research on this topic than the others?

It is possible to answer this question by analyzing several factors, including collaboration networks, most relevant authors, and author production over time. This is in accordance with the analysis mentioned above. Archarya, Bangsod, Li, and Mizra are some of the key contributors to the IoT LWC research field. Archarya. B played a significant role in this research field as he was identified as the key researcher by a number of analysis tools in this research.

- Does this topic pertain to any trending topics? If so, what are they?

According to the analysis of trend topics, thematic map, and factorial analysis, IoT and IoT lightweight cryptography can be identified as some of the trending research topics in the field of IoT. Raspberry Pi is another trending topic identified by the thematic map in this research.

- Which are the leading journals that publish research on this topic.?

Although *IEEE Access* is the leading journal for IoT LWC, *Lecture Notes on Computer Science* and *Lecture Notes on Electronics Engineering* are also significant contributors to the

field. Another important journal on this topic is the *IEEE Internet of Things Journal*. MDPI's *Sensors* is also another important journal that was identified by the most relevant journal analysis as the number four contributor.

- What type of co-relations exist between researchers in terms of their research?

According to the analysis of the data, there are certain clusters of co-relations that can be identified, such as the collaboration network. A total of 10 clusters have been identified, and 2 of them are leading research clusters led by Archarya and Bansod and another cluster with 9 researchers but no leader. It has also been reported that there are a few small clusters of members with two to six members as well.

- What are the relationships between keywords used in articles?

Internet of Things, lightweight cryptography, security, cryptography, encryption, and authentication are some of the leading keywords used by authors as indicated by the word cloud, the co-word net, and the most relevant words based on word frequency over time. It is clear from the list of keywords that they have a strong relationship with each other.

### 5.2. Limitations of the Study

This study analyzed the bibliometric data for LWC successfully, but there are some limitations that can be improved in future studies. Our first limitation was that we chose only three databases, but other databases containing more articles are available as well. Despite this limitation, we have selected the largest and most relevant databases to mitigate it. Additionally, only English-language articles were selected, even though other languages might have been published as well. Most articles are published in English, which reduces this limitation significantly. By selecting articles from 2015 to 2023, we may have excluded articles published prior to that period. Due to the relatively new nature of the subject area, we wanted to limit the study to the latest and most relevant articles.

### 6. Conclusions and Future Research Directions

In conclusion, IoT lightweight cryptography presents many opportunities for researchers in an emerging field and is a rapidly growing research area. In addition to device-based encryption, project-based encryption, and equipment-based encryption research could also be expanded to various types of lightweight encryption applications. A device-based analysis is of particular importance since many established devices, such as Raspberry Pi, Arduino, and Beegle, require lightweight encryption. As we mentioned in the Introduction, Raspberry Pi is a board type that has many advantages and is gaining a large share of the market rapidly [53]. Currently, Raspberry Pi does not offer such lightweight encryption. Considering the unique limitations and rapid development of Raspberry Pi devices, it is crucial to have tailor-made encryption. Developing lightweight algorithms specifically for Raspberry Pi is a positive contribution to the technology. In the future, we plan to develop a lightweight encryption algorithm for Raspberry Pi to be compatible with its specifications. Algorithms such as KATAN [36], Blowfish [54], the Data Encryption Standard (DES) [55], and the Advanced Encryption Standard (AES) [56] can also be considered as base algorithms for developing a new lightweight encryption algorithm. Researchers will be able to come up with this kind of research based on the above analysis, which will contribute enormously to the IoT industry in the future.

**Author Contributions:** All authors contributed to the study conception and design. Material preparation, data collection, and analysis were performed by Z.D. and B.S. The first draft of the manuscript was written by Z.D. and B.S. All authors commented on previous versions of the manuscript. S.A. reviewed the article and S.T. contributed at the supervisory level to the article. All authors have read and agreed to the published version of the manuscript.

**Funding:** This research received no external funding.

**Data Availability Statement:** Source files that were used to analyze data are available on the Figshare repository: https://doi.org/10.6084/m9.figshare.24434035.v1 (accessed on 15 October 2023).

**Conflicts of Interest:** The Authors declare no conflict of interest.

**Abbreviations**

The following abbreviations are used in this manuscript:

| | |
|---|---|
| IoT | Internet of Things |
| LWC | Lightweight cryptography |
| RA | Research article |
| WOS | Web of Science |
| OT | Over time |
| BR | Bibliometric review |
| SBC | Single-board computers |
| HDMI | High-definition multimedia interface. |
| CSI | Camera serial interface |
| DSI | Display serial interface |
| WLAN | Wireless local area network |
| IEEE | Institute of Electronics and Electrical Engineers |
| NIST | National Institute of Standards and Technology. |
| HLA | Hybrid lightweight algorithm |
| IoT LWC | Internet of Things Lightweight Cryptography |
| FPGA | Field-programmable gate array |

**Appendix A. Article Search**

Scopus article search method

1. Load the Scopus database to the browser.
2. There are many criteria available for "search within", but in this case, we selected "keywords" to filter the most relevant article set as per Figure A1 and filter by time range as per Figure A2 limits the article search to the most recent article set.
3. Proceed to search for other word combinations.

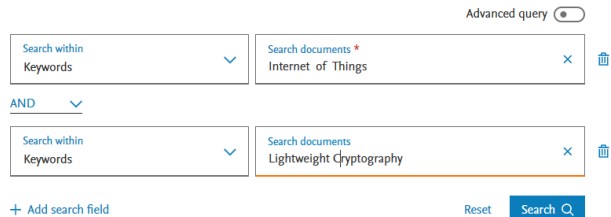

**Figure A1.** Scopus article search interface.

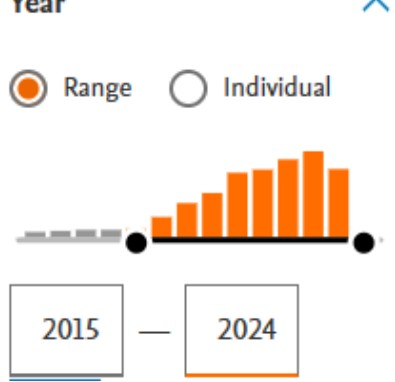

**Figure A2.** Scopus: time period filter.

4. Conduct an article search for other word combinations.

Follow the same procedure for IEEE Xplore as per Figure A4 and for WOS as per Figure A3.

The following figures provide the idea of search interfaces and keyword entry methods.

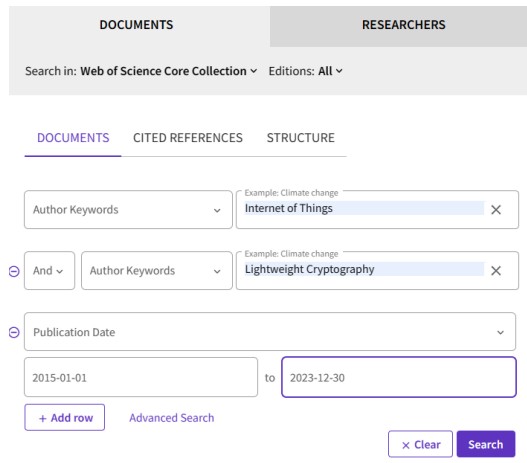

**Figure A3.** WOS article search interface.

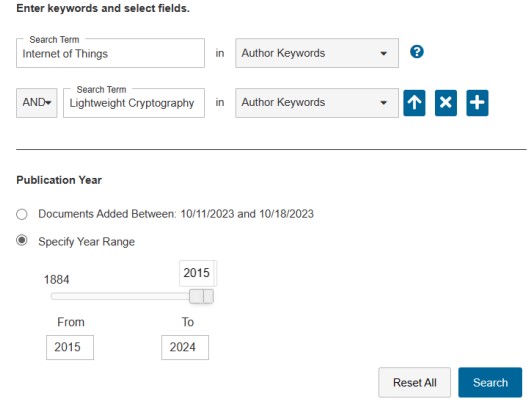

**Figure A4.** IEEE Xplore article search interface.

### Appendix B. Data Integration and Cleansing

Integrate Files belongs to the Scopus Database.

1.  Open Microsoft Excel and open a new Workbook.
    Home > Select Blank Workbook.
2.  Load files into the Workbook.
    Data > From CSV/TEXT > Import Data Window > Select File > Load > Load/SAVE
    (Import all the files belonging to Scopus using the same method.)
3.  Files can be appended using the below method.
    Query > Append > Select 3 or more Table to Append > Select files from the list > OK
    Data Cleansing for Scopus Database.

1.  Open the previously appended file on the screen.
2.  Follow the below procedure.
    Table Design > Remove Duplicate > Select topic as criteria.

IEEE database files are also prepared as the above method, but WOS files are in .xlsx format. The same procedure for loading files could be run but extra selection is needed as it converts the data set to table format. The other two appended files were copied and pasted to the suitable columns to prepare the final file for analysis. Data cleansing was performed as per the above data cleansing method.

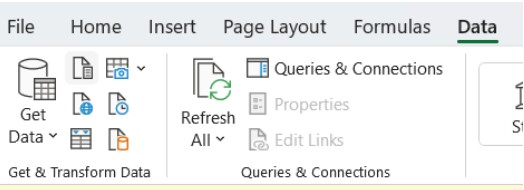

**Figure A5.** Load data from file.

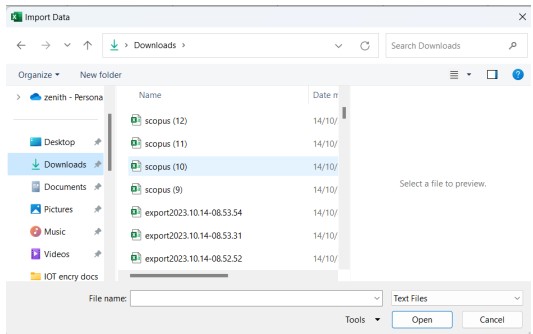

**Figure A6.** Import data.

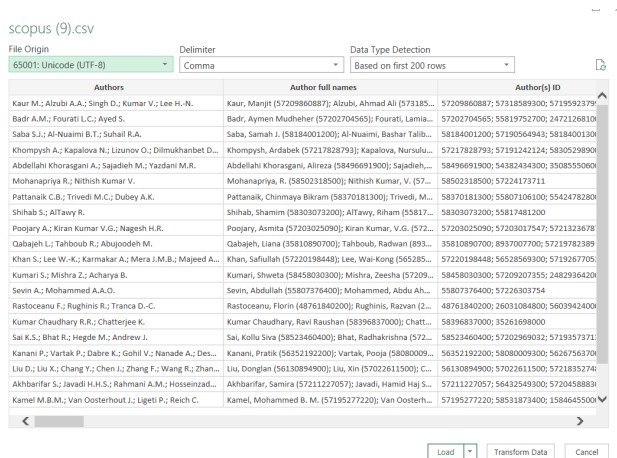

**Figure A7.** Load data.

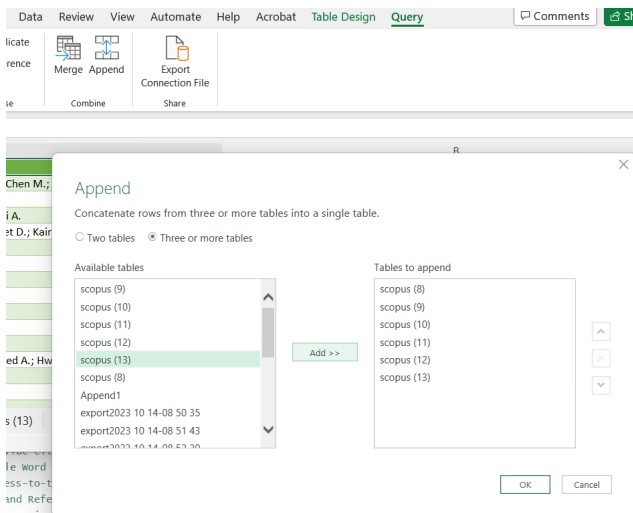

**Figure A8.** Append data.

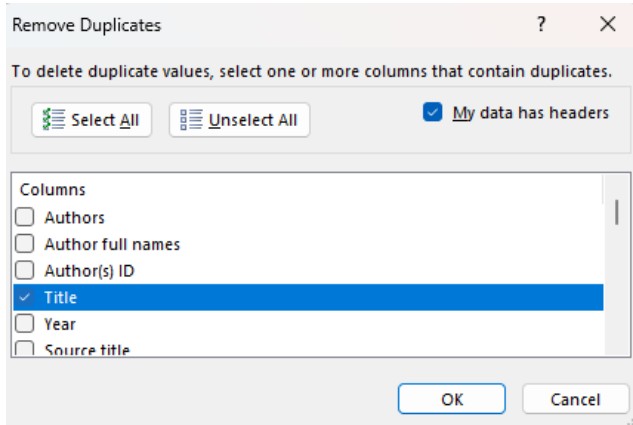

**Figure A9.** Remove duplicates.

The process in Figure A9 will help to remove duplicates but sort titles according to alphabetical order and checking the data set manually will help to validate whether all duplicates were removed or not.

**Appendix C. Data Analysis**

Data Analysis Using R Package.

R Package can be downloaded from https://cran.r-project.org (accessed on 15 October 2023)

1.  Download and install R Package.
2.  Open R Consol, and install the bibliometrix library in the following way.
    Use this command in Command Prompt: install.packages("bibliometrix")
    Follow further instructions to install.
3.  Open biblioshiny
    Use these commands on the prompt after installation:
    >Library (bibliometrix);
    >biblioshiny().
4.  It should open the interface as per Figure A11
5.  Loading the dataset set in CSV format is another important step. Selected options as per Figure A12
6.  Run required reports as per the example in the figure.
    Select the Options button, adjust the option, and press the Start button to provide results.
7.  Download results.
8.  Repeat the procedure for other required analysis options.

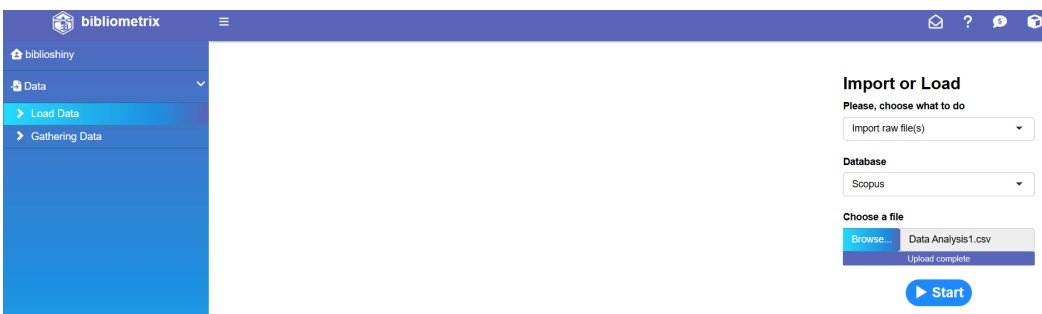

**Figure A10.** Load data from file.

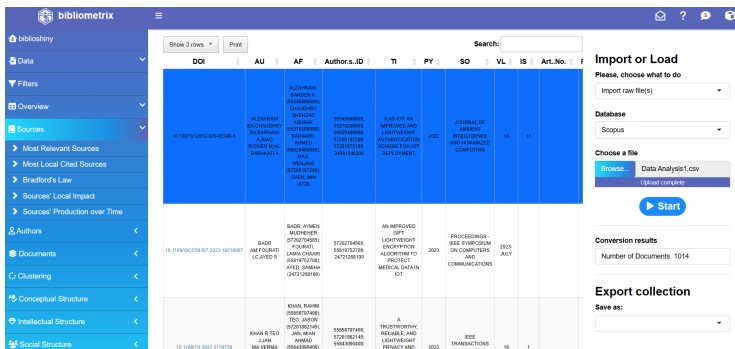

**Figure A11.** Analysis options.

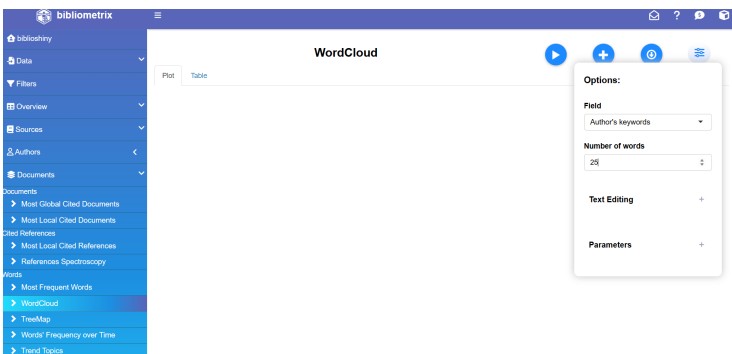

**Figure A12.** Example of analysis.

VOS Viewer Analysis.

1.  Install VOSviewer from this URL: https://www.vosviewer.com/download (accessed on 15 October 2023)
2.  Open VOS Viewer, and follow the below steps to create analysis output.
    Select Create > Create a Map based on Bibliometric Data > Read Data from bibliometric data files > Select Scopus > Load the file > Select required analysis combination > Select Finish.
    The analysis could be obtained as network, overlay, or density visualization.
    These results could be obtained as screenshots or direct downloads.
    The following screenshots provide a better understanding of the process.

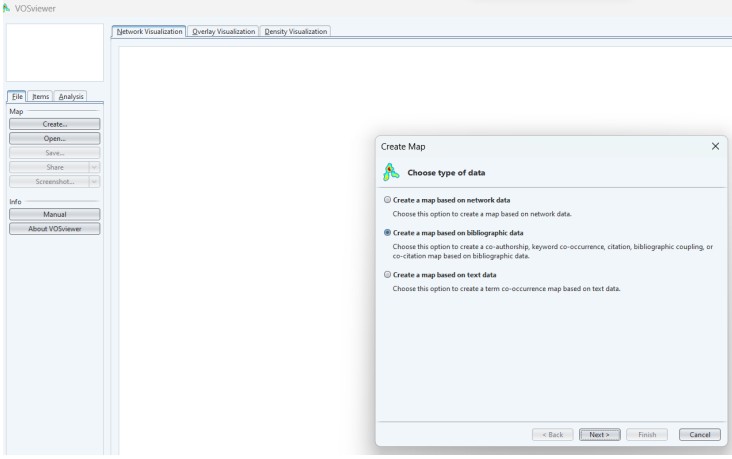

**Figure A13.** Starting to create analysis.

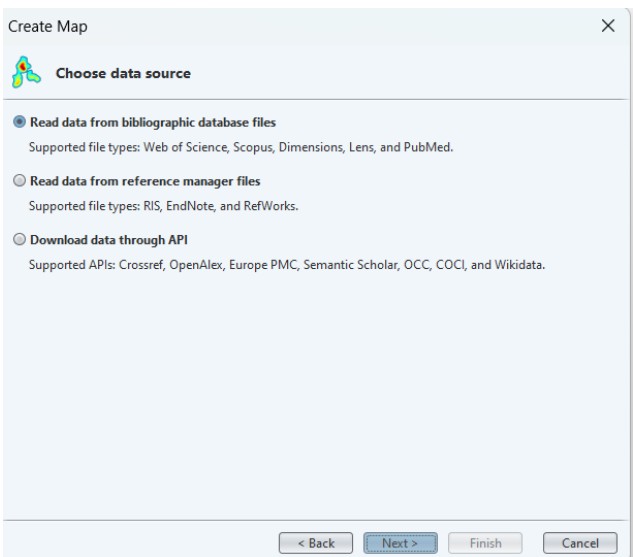

**Figure A14.** Analysis options.

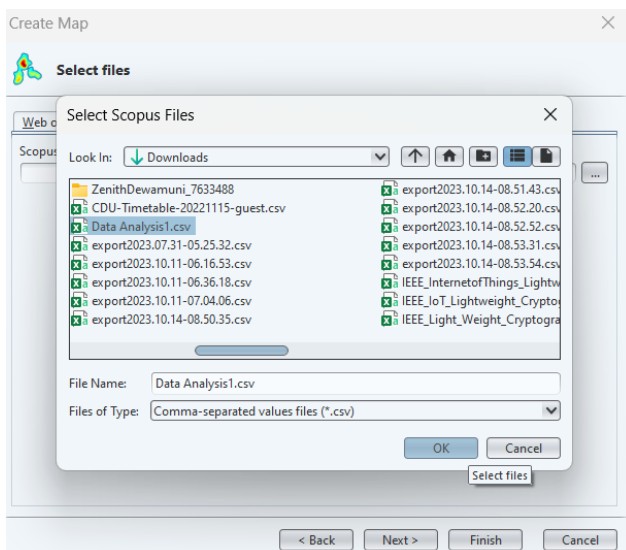

**Figure A15.** File selection for analysis.

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
