# Peer review of "Bibliometric Analysis of IoT Lightweight Cryptography"

_information, doi:10.3390/info14120635_

Round 1

Reviewer 1 Report

Comments and Suggestions for Authors

The paper is very interesting and the subject matter is very topical. However, there are certain shortcomings and aspects that could be improved.

Lines 41-43: It is not clear from this text that family Ascon was selected among several algorithms in the NIST standarization process. It seams that the process is still open.

Table 1: the last column (Bibliometrc Review) is not needed because all values are equal.

The biggest error I see in the article is in section 2.2, related to search strings. The result is very poor, since "light weight" is included, but not "lightweight" or "light-weight", which are written as such throughout the document. It would also be interesting to consider publications that include "Internet of Things" and not just IoT. With these words, the search result is different.

This shortcoming is sufficient for the rest of the article not to be considered for publication, as one of the most important aspects of a search is to know what you are looking for.

On the other hand, although I am not an expert in bibliometric analysis, I do not see the need to repeat the results obtained for the most repeated keywords, i.e. sections 3.2, 3.5 and 3.6 show results about the same thing, albeit with slightly different results.

In Figure 10, Internet of Things appears 3 times in the legend.

Section 3.12, lines 395-396, explains how clusters are drawn. This explanation should be included before the first cluster in the article. 

I hope this helps you to improve your research and I encourage you to resend it.

Author Response

Dear Reviewer,

Thanks for your time, please find the responses attached.

Regards

Bharani

Reviewer 2 Report

Comments and Suggestions for Authors

In this paper, authors  provide an extensive study of IoT light weight cryptography by employing a bibliometric analysis. The work have academic reference significance for researchers to identifying research trends and patterns in Raspberry Pi security.  But I have some comments as follows: 

1. The motivation of this study needs to be further highlighted in the Introduction, preferably with a bulleted or enumerated list.

2.There is too much content in the introduction section, and the sub-section of 1.3 should be treated as a separate section.

3. In table 1, the first row, the year of "2023" should be corrected to "2019" and the Source Journal of "Internet of Things" should be revised to "IEEE INTERNET OF THINGS JOURNAL" . The corresponding information in reference [16] should be corrected.

4. The literature in row 6 of Table 1 cannot be retrieved, and the title and source journal are inconsistent with literature [4].

5. The literature in row 7 of Table 1 cannot be retrieved, and the title and source journal are inconsistent with literature [14].

6. The literature in row 8 of Table 1, the year of "2017" should be corrected to "2022".

7. Figure 1, 2 need to be redrawn, errors include the connection line not in contact with the connected object, and missing lines in the rectangular box.

8. Table 5, 6, 7 and 8 should be changed to three-line tables.

9. Punctuation should not appear at the end of the title, such as the title in section 3.9, 3.10,3.11, 4.1, and 3.11.

10. The font size of the text and numbers in Figures should be adjusted appropriately so that the size of the text in  figures should not exceed the size of the text in the main text. In addition, there are some figures with too large blank lines before and after, which need to be removed.

Comments on the Quality of English Language

The meaning expressed in English in this paper can basically be understood. However, there are some grammar and spelling errors in the paper.

Author Response

Dear Reviewer,

Thanks for your comments and please find the responses attached.

Regards

Bharani

Reviewer 3 Report

Comments and Suggestions for Authors

The paper "Bibliometric analysis of IoT Light Weight Cryptography" tries to identify research trends and patterns in Raspberry Pi security concerning IoT Light Weight Cryptography.

The context and the literature review are presented in the Introduction Chapter along with the paper goals.

Raspberry Pi is introduced in 1.1. Scratch is a software that is not " built into the Raspberry Pi", more like offered as recommended software of an operating system. Also, there is a typo in the "IEEE 8.2.011 Wireless LAN," and in newer versions there is also USB 3.0 not only 2.0

In 1.2 Challenges of IoT Lightweight Cryptographic Algorithms are presented. However, the authors contradict themselves with: “In comparison with traditional cryptographic algorithms, lightweight algorithms are more resource- and power -intensive" I think the contrary is true.

In 1.3 the literature review is made. Mainly review and overview papers are targeted and the focus is on the current lack of bibliometric analysis in the IoT Lightweight Cryptography papers, which is a valid point.

Chapter 2 presents the Materials and Methods with figure 1 illustrating the basic method of bibliometric analysis on the Scopus, WoS and IEEE articles. The results are presented in Chapter 3. They are extensive and the figures included are suggestive.

The results discussion is presented in Chapter 4. The study limitations should be discussed like articles that are not in English language. Also, some validation methods should be discussed for example testing that articles with specific keywords actually focus on that subject in their content.

Finally, the role of Raspberry Pi in the paper content is much less than described in the abstract. For example, in Chapter 2.2 Raspberry Pi as a keyword was not presented/discussed. Either it should be better integrated in the analysis or sidelined.

Comments on the Quality of English Language

Good use of the English Language.

Author Response

(The authors gave the same response as above.)

Reviewer 4 Report

Comments and Suggestions for Authors

The paper titled "Blending Artificial Intelligence and Machine Learning with the Internet of Things: Emerging Trends, Issues, and Challenges" presents an interesting analysis of research trends and patterns in Raspberry Pi security within the context of the Internet of Things (IoT). However, there are some areas where the paper could be improved:

  1. Clarity and Structure:

    • Consider revising the title to accurately reflect the paper's scope.
    • The abstract provides an overview of the paper's findings, but it would be helpful to briefly mention the methodology used in the analysis.
  2. Methodology:

    • The paper mentions the use of R, VOS viewer, and the bibliometric library for analysis but lacks details on the specific methods and parameters used. Providing more information on the methodology would enhance the paper's reproducibility.
  3. Data Sources:

    • Mentioning the publication date range of the articles included in the analysis would provide context to readers. IoT research evolves rapidly, and older articles may not reflect current trends and challenges.
  4. Analysis Findings:

    • While the paper mentions that "India is the leading research country," it would be helpful to explain why this is the case. Are there specific factors contributing to India's prominence in this area?
    • The paper identifies "Archarya.B" and "Bansod.G" as relevant authors. Providing brief information about their contributions or areas of expertise would add value to the analysis.
    • The mention of "Internet of Things, light-weight cryptography, and cryptography" as the most relevant sets of words could be further elaborated. What specific trends or patterns within these keywords were identified?
    • The significance of "IEEE Access" as the most significant journal should be explained. Is it because of the number of articles, the quality of research, or some other factor?
  5. Future Directions:

    • The paper suggests that developing a lightweight cryptographic algorithm for Raspberry Pi boards would be a significant future research focus. It would be beneficial to provide more insight into why this is a critical area for future research and what potential challenges researchers might face.
  6. References:

    • Ensure that the paper includes a comprehensive list of references to support the claims and findings presented.
    • Visualizations:
    • Consider including visualizations from the bibliometric analysis to provide readers with a better understanding of the trends and patterns identified.
Comments on the Quality of English Language

Language and Grammar:

  • Review the paper for language and grammar errors to improve clarity and readability.

Author Response

(The authors gave the same response as above.)

Round 2

Reviewer 1 Report

Comments and Suggestions for Authors

The paper can be published in its current form.

Author Response

Thanks for accepting the corrections.

Reviewer 3 Report

Comments and Suggestions for Authors

All signaled problems were handled by the authors.

Author Response

Thanks for accepting the comments.

Reviewer 4 Report

Comments and Suggestions for Authors

The paper provides valuable insights into the field of IoT Light Weight Cryptography, and the use of bibliometric analysis is commendable.

Author Response

Thanks for accepting the changes.